# Anon: Exploring the Adaptivity of Optimizers and Beyond

## Abstract

Adaptive optimizers such as Adam have achieved great success in training large-scale models like large language models and diffusion models. However, they often generalize worse than non-adaptive methods, such as SGD on classical architectures like CNNs. We identify a key cause of this performance gap: *adaptivity* in pre-conditioners, which limits the optimizer's ability to adapt to diverse optimization landscapes. To address this, we propose **Anon** (**A**daptivity **N**on-restricted **O**ptimizer with **N**ovel convergence technique), a novel optimizer with **continuously tunable adaptivity** $\gamma \in \mathbb{R}$, allowing it to interpolate between SGD-like and Adam-like behaviors and even extrapolate beyond both. To ensure convergence across the entire adaptivity spectrum, we introduce *incremental delay update (IDU)*, a novel mechanism that is more flexible than AMSGrad's hard max-tracking strategy and enhances robustness to gradient noise. We theoretically establish convergence guarantees under both convex and non-convex settings. Empirically, Anon consistently outperforms state-of-the-art optimizers on representative image classification, diffusion, and language modeling tasks. These results demonstrate that adaptivity can serve as a valuable tunable design principle, and Anon provides the first unified and reliable framework capable of bridging the gap between classical and modern optimizers and surpassing their advantageous properties. Our code is available at https://anonymous.4open.science/r/Anon-6511/.

## 1. Introduction

Modern deep learning models rely heavily on optimization algorithms for effective training. Despite the wide success of adaptive optimizers such as Adam (Kingma & Ba, 2014) in large-scale models like diffusion networks (Nichol & Dhariwal, 2021; Rombach et al., 2022) and large language models (LLMs) (Brown et al., 2020; Touvron et al., 2023), they are often outperformed by non-adaptive methods such as SGD (Robbins & Monro, 1951) in classical architectures like CNNs (Wilson et al., 2017). These discrepancies raise a critical question: *Why do existing optimizers fail to generalize across diverse model families?*

We identify a key cause of this performance gap as ***adaptivity*** in pre-conditioners (i.e., the matrix that rescales the gradient before the step; SGD uses the identity, while Adam uses a data-dependent diagonal matrix). Whereas SGD applies fixed step sizes, adaptive optimizers such as Adam scale updates by gradient statistics, implicitly encoding an adaptivity level $A$ throughout training. This $A$, fixed without considering task-specific gradient distributions, can create a mismatch between the optimizer's adaptivity and the task's optimization landscape, potentially degrading generalization performance and rendering optimizers overly specialized. This motivates us to formalize and analyze adaptivity as a first-class property of optimizers.

To address this, we introduce a unified view of adaptivity, defined as the log-sensitivity of the pre-conditioner to global gradient scaling (§2.2). Existing optimizers correspond to fixed points on this adaptivity spectrum: SGD ($A = 0$), RMSProp (Graves, 2013) ($A \approx 1$), and Adam ($A \approx 1$). However, no method supports continuous control across $A \in \mathbb{R}$ with guaranteed stability.

We propose **Anon**, an **A**daptivity **N**on-restricted **O**ptimizer with **N**ovel convergence technique that enables *real-valued, tunable adaptivity* via a hyperparameter $\gamma \in \mathbb{R}$. Anon interpolates between SGD-like and Adam-like updates and even extrapolates beyond them. We note that such adaptivity comes with an important tradeoff: extreme adaptivity (e.g., $\gamma < 0$ or $\gamma > 1$) risks instability and divergence. To tackle this tradeoff, we design a new convergence technique named **incremental delay update (IDU)**, which replaces hard max-tracking (e.g., in AMSGrad) with a soft, multi-scale accumulator that is provably stable.

**Our contributions are as follows**:

- We define a formal notion of adaptivity as a continuous

[1]Anonymous Institution, Anonymous City, Anonymous Region, Anonymous Country. Correspondence to: Anonymous Author <anon.email@domain.com>.

Preliminary work. Under review by the International Conference on Machine Learning (ICML). Do not distribute.

control variable that unifies SGD, Adam, and beyond, offering a unifying lens to guide the design of future optimizers (§2.2).

- Through our analysis, we propose **Anon**, a novel universal optimizer which has tunable adaptivity. Anon's extensive range of adaptivity and adjustment endows the optimizer with the capability to surpass the performance ceiling inherent in previous optimizers (§3.1).

- We propose a novel technique named *incremental delay update*, which eliminates the non-convergence risks in Anon arising from excessive range of adaptivity adjustment and anomalous negative adaptivity. We theoretically establish the convergence of Anon in both online convex and non-convex stochastic settings. In addition, we show that IDU can address convergence issues more effectively than AMSGrad's max-tracking approach (§3.3).

- We conduct extensive experiments in image classification, language, and generative modeling, where Anon consistently outperforms strong baselines across tasks and architectures (§4).

This work advocates for viewing adaptivity as a tunable principle and delivers the first provably stable, unified optimization framework that spans the full adaptivity spectrum.

## 2. Preliminaries

### 2.1. Review of the Frame of Optimizers

We focus on first-order optimizers, which are widely used to train deep learning models. To facilitate a unified understanding of their differences and commonalities, we introduce a generic framework, summarized in Algorithm 1.

---
**Algorithm 1:** Generic Optimizer Method Frame
---
1 Input: $\boldsymbol{\theta}, \eta, \{\phi_t, \psi_t\}_{t=1}^{\infty}$
2 **while** $\boldsymbol{\theta}_t$ not converged **do**
3    $\boldsymbol{g}_t \leftarrow \nabla f_t(\boldsymbol{\theta}_t)$
4    $\boldsymbol{m}_t \leftarrow (\phi_t(\boldsymbol{g}_{1:t,1}), ..., \phi_t(\boldsymbol{g}_{1:t,d}))^{\top}$
5    $S_t \leftarrow \text{diag}(\psi_t(\boldsymbol{g}_{1:t,1}), ..., \psi_t(\boldsymbol{g}_{1:t,d}))$
6    $\boldsymbol{\theta}_t \leftarrow \Pi_{\mathcal{F}, S_t}(\boldsymbol{\theta}_{t-1} - \eta(t) S_t^{-1} \boldsymbol{m}_t)$
7 **end while**

---

Here, $\mathcal{F}$ denotes the convex feasible set. $\boldsymbol{\theta} \in \mathcal{F}$ is the parameter to be optimal. Define $f(\boldsymbol{\theta})$ as a vector-valued function to minimize. $S_t$ is a diagonal matrix where $S_{t,i,i} := \psi_t(\boldsymbol{g}_{1:t,i})$. $\psi_t$ is the pre-conditioner function. $\prod_{\mathcal{F},S}(y) = \text{argmin}_{x \in \mathcal{F}} \|S^{1/2}(x - y)\|$ denotes the projection of $y$ onto $\mathcal{F}$ under the scaling matrix $S$. The scheduler $\eta$ controls the learning rate at each step, which can be constant or scheduled via strategies such as cosine annealing (Loshchilov & Hutter, 2016). $\boldsymbol{g}_t$ is the gradient at step $t$. $\boldsymbol{m}_t$ is a vector where $\boldsymbol{m}_{t,i} := \phi_t(\boldsymbol{g}_{1:t,i})$. The momentum operator $\phi_t : \mathbb{R}^t \to \mathbb{R}$ is typically implemented as a moving average of past gradients. The two common variants are:

$$EMA(\boldsymbol{x}_{1:t}; \beta) = \frac{1-\beta}{(1-\beta^t)} \sum_{i=1}^{t} \beta^{t-i} x_i, \qquad (1)$$

$$M(\boldsymbol{x}_{1:t}; \beta) = \sum_{i=1}^{t} \beta^{t-i} x_i, \qquad (2)$$

where *EMA* denotes the exponential moving average with bias correction. *M* refers to the classical momentum without normalization. Both operators serve to smooth the gradient history.

**Since the smoothing behavior of $\phi$ is similar across optimizers, the key differentiator lies in the design of the pre-conditioner $\psi$, which directly modulates the effective step size and direction for each parameter.** Thus, we focus our subsequent analysis on the properties and effects of $\psi$. While the momentum functions $\phi_t$ are largely similar across optimizers, primarily serving to dampen oscillations and accelerate progress in consistent gradient directions, the pre-conditioner functions $\psi_t : \mathbb{R}^t \to \mathbb{R}_+$ differ significantly and play a crucial role in shaping the optimizer's behavior, such as its convergence rate, stability, and adaptability to heterogeneous parameter landscapes. We summarize the designs of $\phi$ and $\psi$ for representative optimizers in Table 1.

As shown in Table 1, the momentum components $\phi$ exhibit similar behaviors across different optimizers. This observation indicates that the key distinction among optimizers originates from the design of $\psi$, rather than $\phi$. In fact, if the bias correction factor $1/(1-\beta^t)$ is omitted in Exponential Moving Average (EMA), it essentially reduces to a classical momentum $M$ scaled by a constant coefficient $1 - \beta$. Therefore, for the remainder of this paper, we will primarily focus on analyzing the characteristics of the preconditioner $\psi$, assuming a shared momentum $\phi$ across all optimizers unless otherwise specified.

Extensive empirical evidence has shown that SGD and SGDM often achieve better generalization than Adam in classical architectures such as ResNet (He et al., 2016), whereas Adam typically outperforms SGD in more complex architectures such as transformers. Understanding the fundamental causes behind this divergence remains an important question, with significant implications for the development of more effective optimizers. Several hypotheses have been proposed, including that Adam can escape saddle points more efficiently than SGD (Staib et al., 2019), and that SGD tends to find flatter minima whereas Adam is biased toward sharper minima, leading to superior generalization for SGD (Wilson et al., 2017). Regardless of the specific explanations, we hypothesize that the ultimate cause lies in how optimizers scale the loss landscape, a property we refer to as adaptivity. **However, a unified measure to quantify this behavior is currently lacking. To address this, we explicitly design a formal adaptivity metric, which is necessary**

*Table 1.* Summary of momentum functions and pre-conditioners for representative optimizers (Polyak, 1964; Luo et al., 2019; Zhuang et al., 2020). For full expressions of complex terms ($A_t^{\text{AMSGrad}}$, $A_t^{\text{AdaBound}}$, $A_t^{\text{AdaBelief}}$, $A_t^{\text{Anon}}$), please refer to Table 6 of Appendix B.1.

| OPTIMIZER | $\phi_t(x)$ | $\psi_t(x)$ | $A_t(\psi, x)$ |
|---|---|---|---|
| SGD | $x_t$ | $1$ | $0$ |
| SGDM | $M(x; \beta)$ | $1$ | $0$ |
| RMSPROP | $x_t$ | $\sqrt{EMA(x^2; \beta_2)} + \epsilon$ | $\frac{1}{1 + \epsilon/\sqrt{EMA(x^2; \beta_2)}}$ $(\approx 1)$ |
| ADAM | $EMA(x; \beta_1)$ | $\psi_t^{\text{RMSPROP}}$ | $A_t^{\text{RMSPROP}}$ |
| AMSGRAD | $\phi_t^{\text{ADAM}}$ | $\max_{i \in [t]}\{\psi_i^{\text{RMSPROP}}\}$ | $[0, 1]\ (\approx 1)$ |
| ADABOUND | $\phi_t^{\text{ADAM}}$ | $\text{CLIP}(\psi_t^{\text{RMSPROP}}, f_l(t), f_u(t))$ | $[0, 1]$ $(1 \to 0)$ |
| ADABELIEF | $\phi_t^{\text{ADAM}}$ | $\left(EMA((x - \phi^{\text{ADAM}})^2 + \epsilon/(1 - \beta_2); \beta_2)\right)^{1/2} + \epsilon$ | $[0, 1]\ (\approx 1)$ |
| ANON | $\phi_t^{\text{ADAM}}$ | $\psi_t^{\text{ANON}}$ ((6)) | $\approx \gamma$ |

**to mathematically distinguish the scaling mechanisms of SGD and Adam to guide the design of extrapolative optimizers.** We will study how adaptivity affects optimization in § 3.2. Before that, we first give a formal definition of adaptivity.

### 2.2. The Adaptivity of Existing Optimizers

We formalize the concept of adaptivity based on the framework described in Algorithm 1.

**Definition 2.1.** Suppose the pre-conditioner $\psi_n$ is continuous. For any optimizer following Algorithm 1, we define the adaptivity $A$ of its pre-conditioner $\psi$ as

$$A_n(\psi, x_{1:n}) = \nabla_k \ln \psi_n(k x_{1:n})\big|_{k=1}. \tag{3}$$

Furthermore, we define two pre-conditioners $\psi$ and $\psi'$ are equivalent if and only if $A_n(\psi, x_{1:n}) = A_n(\psi', x_{1:n})$ for all $x_{1:n} \in \mathbb{R}^n$ and $n \in \mathbb{N}_+$.

**Intuition Explanation** $A$ is a measure of the pre-conditioner's "response" to changes in the gradient's scale. $A = 0$ (like SGD) means the pre-conditioner is "non-reactive" and completely ignores the overall gradient scale. $A \approx 1$ (like Adam) means the pre-conditioner is "compensatory", adjusting itself with a strength of 1 to offset changes in gradient scale. Notably, according to Definition 2.1, the adaptivity $A$ depends not only on the functional form of $\psi$, but also on the sequence of historical gradients $g_{1:t}$. This dependence reflects the fact that pre-conditioning is inherently dynamic: even for a fixed $\psi$, its adaptivity can vary during training as the distribution of gradients evolves. Separately, we introduce an important equivalence notion between pre-conditioners: even if two optimizers use different $\psi$ functions, they may be essentially equivalent from an adaptivity perspective.

**Theorem 2.2.** *If $\psi$ and $\psi'$ are from the same equivalence class, there exists $f : \mathbb{N}_+ \to \mathbb{R}_+$ such that $\psi_n(x_{1:n}) = \psi'_n(x_{1:n})f(n)$ for any $x_{1:n} \in \mathbb{R}^n$ and $n \in \mathbb{N}_+$.*

**Decoupling from Scheduler** Theorem 2.2 shows that if two pre-conditioners yield the same adaptivity for any input, then they are equivalent. Specifically, if there exists a scheduler adjustment that can eliminate the difference between two pre-conditioners (e.g., $\psi' = k\psi$ corresponds to $\eta'(t) = k\eta(t)$), we regard them as equivalent strategies. The proof of Theorem 2.2 is deferred to Appendix.

Based on these definitions, we can characterize the adaptivity of several widely used optimizers: For SGD(M), the adaptivity is $A = 0$ in all dimensions, indicating no explicit scaling of the loss landscape. In contrast, for Adam and its variants (e.g., RMSProp, AdaBelief), the adaptivity is approximately $A = 1$, as the contribution of the small $\epsilon$ term is negligible compared to the accumulated gradient statistics most of the time. A more intricate case is AdaBound (Luo et al., 2019), whose adaptivity transitions dynamically from $A \approx 1$ toward $A \approx 0$ as training proceeds. Specifically, AdaBound clamps the pre-conditioner $\psi_t$ between shrinking bounds $\eta_l(t)$ and $\eta_u(t)$:

$$A_t(\psi^{\text{AdaBound}}, x) = \begin{cases} A_t^{\text{RMSProp}}, & \text{if } \eta_l < \psi_t^{\text{RMS}} < \eta_u, \\ 0, & \text{otherwise.} \end{cases} \tag{4}$$

As the bounds tighten over time, AdaBound behaves increasingly like SGD. This is supported by both evidence from Zhuang et al. (2020) and our experiments (Table 5), which indicates that AdaBound struggles in tasks such as GAN and diffusion model training, where high adaptivity is critical. These observations suggest the following: Optimizers with $A = 0$ (e.g., SGD) tend to generalize better on classical architectures such as CNNs, while those with $A = 1$ (e.g., Adam) perform better in complex modern architectures. However, whether $A = 0$, $A = 1$, or other values yield better performance remains an open question, which we explore in the next section.

### 2.3. The Optimal Adaptivity for Tasks

We argue that the optimal adaptivity is inherently task-dependent and often lies beyond the standard range. While

mainstream optimizers typically operate at $A \approx 1$ (Adam) or $A \approx 0$ (SGD), different tasks, ranging from sharp-minima Transformers to flat-minima ResNets, may favor adaptivity values far outside the $[0, 1]$ interval (i.e., $A < 0$ or $A > 1$).

However, as we show in Figure 1, existing methods are theoretically constrained to interpolation. Approaches like Padam (Chen & Gu, 2018) and MADA (Ozkara et al., 2024) attempt to tune adaptivity, but they are fundamentally restricted to the convex hull of SGD and Adam. Extending these methods for extrapolation is non-trivial because the convergence of most adaptive optimizers relies on the critical non-decreasing assumption:

$$\frac{\psi_t(g_{1:t+1,i})}{\eta(t+1)} \geq \frac{\psi_t(g_{1:t,i})}{\eta(t)}, \quad \forall i \in [d], \forall t \in \mathbb{N}_+, \quad (5)$$

which guarantees stability even in worst-case scenarios. Naively constructing an optimizer with negative adaptivity (e.g., via negative powers $\psi = (\psi^{\text{Adam}})^\gamma$ with $\gamma < 0$) typically causes the pre-conditioner value to decrease over time, violating Eq. 5 and leading to divergence. Prior works (Chen & Gu, 2018; Chen et al., 2018) have shown that Padam, when extending adaptivity beyond $[0, 1]$, suffers from divergence in practice. Consequently, current literature lacks a unified framework that can stably extrapolate adaptivity to match the diverse geometric requirements of different tasks without breaking convergence guarantees.

## 3. Extend to All Real Numbers

### 3.1. Adaptivity Tunable Optimizer and Beyond

In §2.2 and §2.3, we have shown that extending adaptivity beyond $[0, 1]$ could be beneficial. However, achieving tunable adaptivity across all real numbers while ensuring convergence remains challenging. We propose a new technique called *incremental delay update (IDU)*, which can ensure the convergence of an optimizer regardless of the value of its adaptivity. We will elaborate the technique in §3.3. Leveraging this technique, we design a novel optimizer *Anon* (**A**daptivity **N**on-restricted **O**ptimizer with **N**ovel convergence technique) with tunable adaptivity and extend the allowable range of adaptivity to all real numbers.

$$\psi_t^{\text{Anon}}(\boldsymbol{x}) = \left( \sum_{j=1}^{\tilde{a}_t} \beta_3^{\tilde{a}_t - j} (1 - \beta_3 \mathbb{I}_{j>1}) \right.$$
$$\left. \cdot EMA^\gamma(\boldsymbol{x}_{a_{j-1}+1:a_j}^2 + \epsilon; \beta_2) \right)^{1/2}. \quad (6)$$

The pseudocode of Anon is presented in Algorithm 2, and all the operations are element-wise. Here, $\widehat{\boldsymbol{m}}_t$ corresponds to $\boldsymbol{m}_t$ in Algorithm 1. $\mathbf{V}_k$ corresponds to $\mathbf{S}_t^{-1}$ in Algorithm 1. $\boldsymbol{s}_t$, $\boldsymbol{\sigma}_k$, $\boldsymbol{v}_k$, and $k$ are intermediate variables. $\gamma$ is a hyperparameter to adjust adaptivity $A$. $\epsilon$ is a small hyperparameter

to avoid division by 0. $\beta_1, \beta_2$ are hyperparameters for *EMA*, $0 \leq \beta_1, \beta_2 < 1$, typically set as 0.9 and 0.999. Let $\{a_n\}$ is a increasing sequence and $a_1 = 1$ (specially, let $a_0 = 0$). Let $\tilde{a}_n = \sum_{i>0} \mathbb{I}_{a_i \leq n}$, so $\tilde{a}_1 = 1$. The pre-conditioner of Anon can be written as (6) ($\beta_3 = 0.5, a_n = 2^{n-1}$).

---

**Algorithm 2:** The Anon Optimizer

---

1   Input: $\eta, \beta_1, \beta_2, \epsilon, \gamma$
2   **Initialize $\boldsymbol{\theta}_0, \boldsymbol{m}_0 \leftarrow \mathbf{0}$, $\boldsymbol{s}_0 \leftarrow \mathbf{0}, t \leftarrow 0, k \leftarrow -1$**
3   **while $\boldsymbol{\theta}_t$ not converged do**
4     $t \leftarrow t + 1$
5     $\boldsymbol{g}_t \leftarrow \nabla f_t(\boldsymbol{\theta}_t)$
6     $\boldsymbol{m}_t \leftarrow \beta_1 \boldsymbol{m}_{t-1} + (1 - \beta_1)\boldsymbol{g}_t$
7     $\widehat{\mathbf{m}}_t \leftarrow \frac{\boldsymbol{m}_t}{1 - \beta_1^t}$
8     $\boldsymbol{s}_t \leftarrow \beta_2 \boldsymbol{s}_{t-1} + (1 - \beta_2)\boldsymbol{g}_t^2$
9     **if $k + 1 = \log_2 t$ do**
10      $k \leftarrow k + 1$
11      $\boldsymbol{\sigma}_k \leftarrow \boldsymbol{s}_t / (1 - \beta_2^{\max(t/2,1)}) + \epsilon$
12      $\boldsymbol{v}_k \leftarrow \sqrt{2/(\frac{1}{\boldsymbol{v}_{k-1}^2} + \boldsymbol{\sigma}_k^\gamma)}$ if $k > 0$ else $\boldsymbol{\sigma}_k^{-\gamma/2}$
13      $\boldsymbol{s}_t \leftarrow \mathbf{0}$
14      $\mathbf{V}_k \leftarrow \text{diag}(v_{k,1}, ..., v_{k,d})$
15     **end if**
16     $\boldsymbol{\theta}_t \leftarrow \Pi_{\mathcal{F}, \mathbf{V}_k^{-1}}(\boldsymbol{\theta}_{t-1} - \eta(t)\mathbf{V}_k \widehat{\boldsymbol{m}}_t)$
17   **end while**

---

**Theorem 3.1.** *For the optimizer Anon described in Algorithm 2, the adaptivity of Anon in $i$-th dimension is $\in [\gamma(1 - k), \gamma)$, where $k = \epsilon / \min_{j \in [\tilde{a}_t]} EMA(\boldsymbol{g}_{a_{j-1}+1:a_j,i}^2; \beta_2)$.*

According to Theorem 3.1, since we also set a small $\epsilon$ by default, we can adjust the adaptivity $A$ of Anon by adjusting the hyperparameter $\gamma$ ($A \approx \gamma$). The proof of Theorem 3.1 is shown in Appendix.

### 3.2. How Adaptivity Influences Behaviors of Optimizers

**Empirical Validations**   To show how adaptivity influences the behaviors of optimizers, we conduct a simple experiment in the loss function $f(x, y) = \ln(1 + \text{Beale}(x, y))/10$, where Beale (Beale, 1955) is a commonly used function to test optimizer performance. We apply appropriate learning rates for SGDM, Adam, AdaBelief, and Anon, and draw the optimization trajectories. We also show the loss landscapes in the view of Anon by scaling the loss landscape according to the pre-conditioner of Anon in epoch 100. The trajectories and loss landscapes after scaling are shown in Figure 2.

**Effect of Scaling**   By changing $\gamma$ from 1.5 to $-0.5$, the adaptivity also changes from 1.5 to $-0.5$ referring to Theorem 3.1. We can find that when $\gamma = 1.5$, Anon takes a shorter path to descend along the y-axis. When $\gamma = 0.5$, the

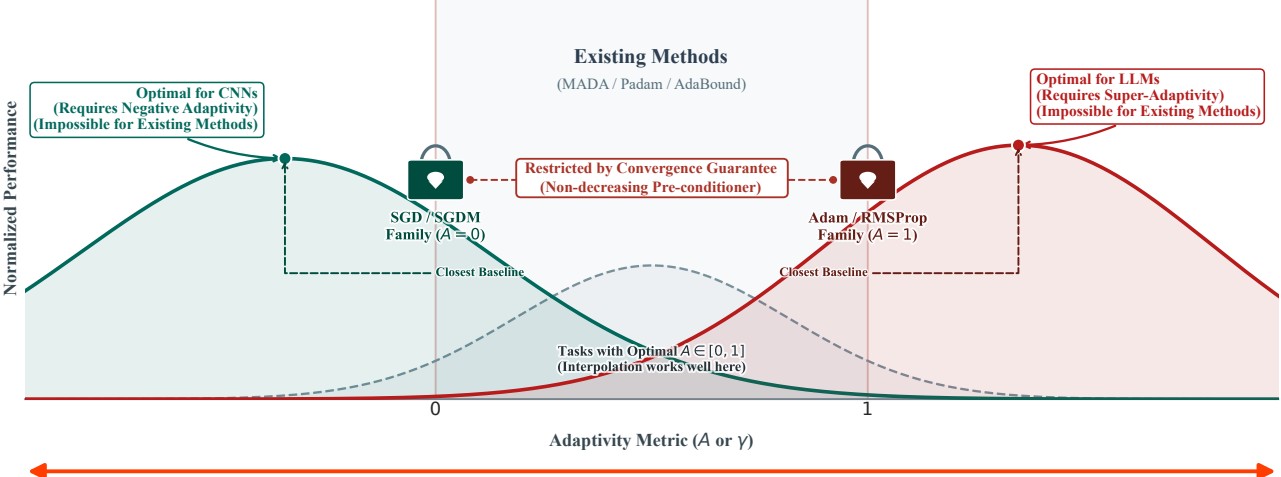

*Figure 1.* Visualizing the Dilemma: The distribution of existing optimizers' adaptivity and the preference of different tasks. Convergence constraints lock existing optimizers in suboptimal regimes.

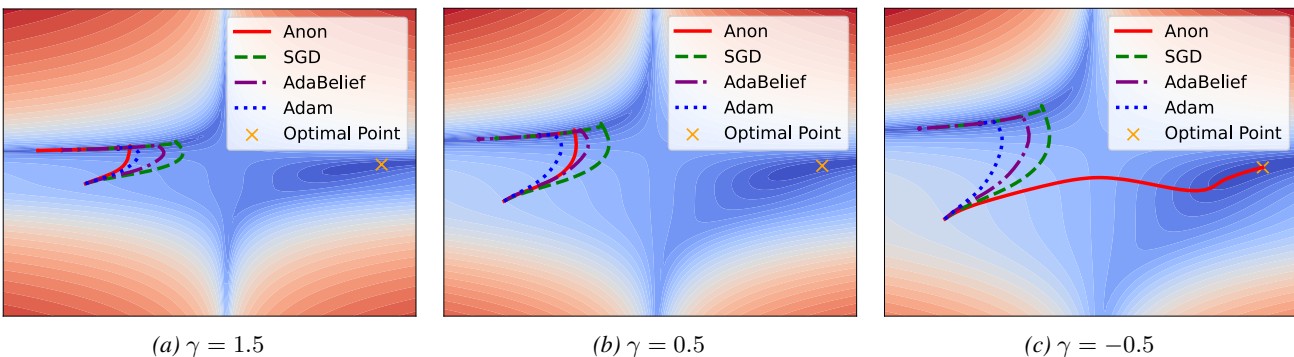

| (a) $\gamma = 1.5$ | (b) $\gamma = 0.5$ | (c) $\gamma = -0.5$ |
|---|---|---|

*Figure 2.* Trajectories of SGDM, Adam, AdaBelief, and Anon. The color change from deep red to deep blue represents the loss from high to low. And the loss landscape displayed is the result of scaling by Anon. More empirical experiments are shown in Appendix B.2 and D.

path is between Adam and SGDM. And when $\gamma = -0.5$, the Anon descends along the x-axis and arrives at the optimal point. We can find that in the progress of $\gamma$'s decreasing, the scale of the x-axis is smaller and smaller than that of the y-axis, so that Anon can choose the right path to reach the optimal point. This example implies that the optimization path of Anon in deep learning training may be greatly different from other optimizers, helping reach a new parameter region that makes the model achieve better performance.

**The Meaning of Negative Adaptivity**  Positive adaptivity typically reduces step sizes for large gradients to help escape saddle points. In contrast, negative adaptivity adopts the opposite strategy by increasing step sizes when gradients are large, which enables the optimizer to escape from sharp minima. Intuitively, higher adaptivity drives the optimization toward steeper minima, whereas lower adaptivity favors flatter regions. **Thus, adaptivity influences the optimizer not only through its path but also by altering its pref-**

erence for specific minima geometries.** This perspective implies that restricting adaptivity to fixed points like A=0 or A=1 is insufficient. Empirically, we find that negative adaptivity is more effective for classical models, while positive adaptivity remains suitable for complex architectures.

### 3.3. Incremental Delay Update

As we state in § 2.3, it is challenging to guarantee the convergence when adaptivity is allowed to take any value. So we propose a new technique *incremental delay update* (IDU), which can be seen as using a new function $U(\boldsymbol{x}; \psi^{\text{old}})$ to replace the old pre-conditioner function $\psi$:

$$U_t(\boldsymbol{x}; \psi_t^{\text{old}}, \{a_n\}, \beta_3) = \left( \sum_{j=1}^{\tilde{a}_t} \beta_3^{\tilde{a}_t - j} \left( 1 - \beta_3 \mathbb{I}_{j>1} \right) \right.$$
$$\left. \cdot \left( \psi_{a_j - a_{j-1}}^{\text{old}} (\boldsymbol{x}_{a_{j-1}+1:a_j}) \right)^2 \right)^{1/2}. \quad (7)$$

Lines 9~15 of Algorithm 2 are the recursive formulas for IDU used in Anon where $\beta_3 = 0.5$, $a_n = 2^{n-1}$ and $\psi^{\text{old}} = EMA^{\gamma}(\mathbf{x}^2 + \epsilon; \beta_2)$. IDU updates the pre-conditioner using accumulated gradient information **only at specific, delayed steps**. This strategy confines unpredictable oscillations within a manageable range, thereby ensuring theoretical convergence while still permitting the pre-conditioner to change non-monotonically. We show the convergence of Anon in Theorem 3.2 (convex cases) and Theorem 3.4 (non-convex cases). And the proofs are provided in Appendix.

**Theorem 3.2.** *(Convergence analysis for online convex optimization) Let $\{\theta_t\}$ and $\{v_k\}$ be the sequence obtained by Algorithm 2, $\gamma \in \mathbb{R}$, $\beta_1 \in [0,1)$, $\beta_2 \in [0,1)$, $\beta_{1,t+1} \in [0, \beta_1]$, $\beta_{1,1} = \beta_1$, $\eta(t) = \frac{\eta_0}{\sqrt{t}}$, for $\forall t \in [T]$. Assume that $\|x - y\|_{\infty} \leq D_{\infty}$ for $\forall x, y \in \mathcal{F}$. Suppose $f(\theta)$ is a convex function, $\|g_t\|_{\infty} \leq G_{\infty}$, for $\forall t \in [T]$, $\theta \in \mathcal{F}$. Let $C_l = \min(G_{\infty}^{-\gamma}, \epsilon^{-\gamma})$, $C_u = \max(G_{\infty}^{-\gamma}, \epsilon^{-\gamma})$, where $\epsilon \in \mathbb{R}_+$ is a very number set in Algorithm 2. The optimal point of $f$ is denoted as $\theta^*$. For $\{\theta_t\}$ generated by Anon, there is a bound on the regret:*

$$\sum_{t=1}^{T}[f_t(\theta_t) - f_t(\theta^*)] \leq \frac{dD_{\infty}^2 c_l^{-1}}{(1-\beta_1)\eta_0} \sum_{k=1}^{\tilde{a}_T - 1} \sqrt{a_{k+1}}$$

$$+ \frac{D_{\infty}^2}{2C_l\eta_1(1-\beta_1)} + \frac{dD_{\infty}G_{\infty}}{1-\beta_1}\sum_{t=1}^{T}\beta_{1,t} + \frac{dG_{\infty}^2 C_u \eta_0}{1-\beta_1}\sqrt{T}$$

$$+ \frac{dD_{\infty}^2 c_l^{-1}}{(1-\beta_1)\eta_0}\sqrt{T} + \sum_{t=1}^{T-1}\frac{\beta_{1,t+1}\mathbb{I}_{\beta_{1,t+1}>\beta_{1,t}}D_{\infty}^2}{2C_l\eta_{t+1}(1-\beta_1)^2} \quad (8)$$

**Corollary 3.3.** *Suppose $\beta_{1,t} = \beta_1\lambda^t$, $0 < \lambda < 1$ in Theorem 3.2, then we have:*

$$\sum_{t=1}^{T}[f_t(\theta_t) - f_t(\theta^*)] \leq \frac{dD_{\infty}^2 c_l^{-1}}{(1-\beta_1)\eta_0}\left(\sqrt{T} + \sum_{k=1}^{\tilde{a}_T - 1}\sqrt{a_{k+1}}\right)$$

$$+ \frac{D_{\infty}^2}{2C_l\eta_1(1-\beta_1)} + \frac{dD_{\infty}G_{\infty}\beta_1}{(1-\beta_1)(1-\lambda)} + \frac{dG_{\infty}^2 C_u \eta_0}{1-\beta_1}\sqrt{T} \quad (9)$$

*It implies the regret of Anon is upper-bounded by $O(\sqrt{T})$ for convex case when $a_n = 2^{n-1}$.*

**Theorem 3.4.** *(Convergence analysis for non-convex stochastic optimization) The update of $\theta_t$ can be described as $\theta_{t+1} = \theta_t - \eta_t V_{\lfloor \log_2 t \rfloor} m_t$, and $m_t = \beta_1 m_{t-1} + (1 - \beta_1)g_t$. Under the assumptions:*

- *$f$ is differentiable and $f^* \leq f \leq F$. $\nabla f(x)$ is L-Lipschitz continuous, i.e. $\|\nabla f(x) - \nabla f(y)\| \leq L\|x - y\|$, $\forall x, y$.*

- *The noisy gradient is unbias and its infinity norm is bounded by $N$, i.e. $\mathbb{E}g_t = \nabla f(x)$, $\|g_t\|_{\infty} \leq N$.*

*The hyperparameters are set as: $\eta_t = \eta_0 t^{-p}$, $\eta_0 > 0$, $p \in (0,1)$ where the bounds are $C_l I \preceq V_{\lfloor \log_2 t \rfloor} \preceq C_u I$, and $0 < C_l < C_u$ ($A \preceq B$ means $B - A$ is a positive semi-definite matrix). And $\epsilon$ and $N$ ensure $C_l$ and $C_u$ exist. For sequence $\{\theta_t\}$ generated by Anon, we have:*

$$\frac{1}{T}\sum_{t=1}^{T}\left\|\nabla f(x_t)\right\|^2 \leq \frac{1}{\eta_0 C_l}T^{p-1}\cdot$$

$$\left(F - f^* + K\int_1^T t^{-2p}\,\mathrm{d}t + J + K\right), \quad (10)$$

*where*

$$J = \frac{\beta_1^2 dN^2}{4L(1-\beta_1)^2} + \frac{3dN^2\eta_0 C_u}{1-\beta_1}\sum_{k=1}^{\tilde{a}_t}(a_k - \mathbb{I}_{k\neq 1})^{-p}, \quad (11)$$

$$K = \left(\frac{1}{1-\beta_1} + \frac{1}{2}\right)L\eta_0^2 N^2 C_u^2 d. \quad (12)$$

Theorem 3.4 shows when $p = 0.5$ and $a_n = 2^{n-1}$, Anon has a convergence rate of $O(\ln T/\sqrt{T})$ for non-convex cases. Note that the convergence rates shown in Theorem 3.2 and Theorem 3.4 are the same as mainstream adaptive optimizers under the strong assumption (5) or using the technique of AMSGrad. And the assumptions and boundedness conditions are standard in the literature and consistent with those adopted in previous works like Luo et al. (2019) and Zhuang et al. (2020).

**Better Noise Robustness.** Other convergence guarantee techniques typically employ alternative methods to ensure (5) holds, thereby guaranteeing optimizer convergence. Noise in the early training stage can greatly influence their performance, making it difficult for these methods to use the information of the latest gradients. As we know, IDU is the first technique that makes optimizers converge and allows (5) to not hold, which will offer Anon (IDU) better noise robustness and flexibility. To evaluate the robustness of IDU against noise, we do further experiments where we compare Anon (IDU) and AMSGrad. Slightly different from the Table 1, AMSGrad is usually implemented in practice in the form: $\max_{i\in[t]}\{\psi_i^{\text{RMSProp}}\sqrt{1-\beta_2^i}\}/\sqrt{1-\beta_2^t}$ (we apply in experiments). But regardless of the first form or the second form, we can extrapolate that AMSGrad's strategy of persistently applying the max operation is highly susceptible to noise interference. We conduct empirical experiments to prove it, and the relevant function settings include:

$$f_t(x) = \begin{cases} 1010x, & \text{if } t \pmod{101} = 1 \\ -10x, & \text{otherwise} \end{cases}, \quad (13)$$

$$N_t = \begin{cases} 500/e^{t-1}, & \text{if } t \pmod 2 = 1 \\ -500/e^{t-1}, & \text{otherwise} \end{cases} \quad (14)$$

with the constraint set $\mathcal{F} = [-1, 1]$. The $f_t(x)$ is the example provided in Reddi et al. (2019), which can make Adam diverge. And $N_t$ is the noise added to the gradients $g_t$. We can observe that the noisy gradient is unbiased and its influence on gradients approaches $0$ with the increase of $t$.

The results of experiments are shown in Figure 3. Note that we set $\gamma = 1$ to make the adaptivity of Anon equivalent to AMSGrad and Adam, and their other hyperparameters are the same. Therefore, we can compare the performances of the two convergence guarantee techniques fairly.

From Figure 3(a)(c), we can see that the regrets divided by $t$ of Anon and AMSGrad approach 0 gradually, meaning they converge. And those of Adam approach a constant, meaning it diverges. Although both Anon and AMSGrad can converge, Figure 3(b)(d) shows that Anon can reach the optimal point $x = -1$ fast, but AMSGrad converges to the optimal point much slower due to the noise, especially when $\beta_2$ is small. The result proves that Anon (IDU) has better noise robustness than AMSGrad, as we have inferred. It forms the theoretical backbone of Anon and opens new avenues for designing flexible optimizers.

## 4. Experiments

In this section, we compare Anon with 13 baseline optimizers, including SGD(M), Adam, AdamW (Loshchilov & Hutter, 2017), Yogi (Zaheer et al., 2018), AdaBound, RAdam (Liu et al., 2019), SWA (Izmailov et al., 2018), Lookahead (Zhang et al., 2019), AdaBelief, Adai (Xie et al., 2022) Lookaround (Zhang et al., 2023), Sophia (Liu et al., 2023), AGD (Yue et al., 2023) and HVAdam (Zhang et al., 2025) by validating Anon in various tasks including image classification tasks on ResNet, image generation on diffusion model and natural language processing tasks on LLMs. Except for experiments on the diffusion model, all the benchmarks are from the data presented in the paper. Therefore, the hyperparameters of other optimizers have been extensively searched.

**Image Classification with CNN**  We conduct experiments on ImageNet (Russakovsky et al., 2015) with ResNet18 and ResNet50. We use the official implementation of AdaBound, AdaBelief and Lookaound, so the replication is exact. For ResNet50, the top-1 accuracy is reported in Table 3. And for ResNet18, the top-1 accuracy is shown in Table 2. We set 1 learning rate for Anon, which corresponds to 0.1 learning rate and 0.9 momentum setting of SGDM, because $EMA(\boldsymbol{x}; 0.9) \approx M(\boldsymbol{x}; 0.9)/10$ according to (2). We set $\gamma = -0.1$ for Anon ($A = -0.1$), and it surpasses the performance of SGDM ($A = 0$). These results prove our guess that the negative adaptivity is more suitable for classical models like CNNs. And the detailed setting is shown in Appendix C.

**Language Modeling**  We train autoregressive models on OpenWebText (Gokaslan & Cohen, 2019) using the official implementation of Sophia (Liu et al., 2023). Our experiments follow the exact experimental setup and hyperpa-

*Table 2.* Top-1 accuracy (%) of ResNet18 on ImageNet. † from Chen & Gu (2018), ‡ from Liu et al. (2019), ∗ from Zhuang et al. (2020).

| Method | Anon | SGDM | AMSGradW |
|---|---|---|---|
| Acc. | **70.06** | 69.94 | 68.78 |
| Method | AdaBelief | AdaBound† | MSVAG∗ |
| Acc. | 69.42 | 68.13 | 68.23 |
| Method | RAdam‡ | Yogi† | Adam‡ |
| Acc. | 67.62 | 66.54 | 65.99 |

*Table 3.* Top-1 accuracy (%) of ResNet50 on ImageNet. † from Xie et al. (2022), ‡ from Zhang et al. (2023), ∗ from Zhang et al. (2025).

| Method | Anon | SGDM | Lookaround | Adam† |
|---|---|---|---|---|
| Acc. | **77.25** | 76.23 | 76.77 | 72.87 |
| Method | Adai† | SWA‡ | Lookahead‡ | HVAdam∗ |
| Acc. | 76.80 | 76.78 | 76.52 | 77.22 |

rameter configurations of Liu et al. (2023). We set $\gamma \geq 1$ and use other optimizers' learning rate setting for Anon. The results of experiments are presented in Table 4, and Anon obtains the lowest validation losses in GPT2-small and GPT2-medium, demonstrating strong performance on LLM training. Note that through our experiments, we find that many variants of Adam are slower than Adam because they introduce extra calculations. But from Table 4 we can see that Anon obtains the compared and even faster speed than Adam. This is because when iterations approach infinity, for the average time cost per iteration, we have

$$E(t^{Adam} - t^{Anon}) \approx t^{vector\text{-}Div} + t^{vector\text{-}Sqrt} - t^{vector\text{-}Mul}$$

$$- C \frac{\log_2 Iters}{Iters} > 0 \; (Iters \to \infty) , \quad (15)$$

and $C$ is the time cost of the operations in line 9~15 of Algorithm 2 per iteration. From (15) we find that the Adam's time cost of per iteration is more than Anon's, since the vector division is slower than vector multiplication. Furthermore, IDU makes the big time cost of vector power operation related to $\gamma \in \mathbb{R}$ in Anon (covered in $C$) approach 0, which greatly improved the practical value of Anon.

*Table 4.* Validation loss and training time on OpenWebText.

| Model | Optimizer | Validation Loss | Time (h) |
|---|---|---|---|
| GPT2-small | Anon$_{\gamma=1.1}$ | **2.93283** | **26.17554** |
| | AdamW | 2.95614 | 26.88118 |
| | Sophia-G | 2.95143 | 28.98702 |
| GPT2-medium | Anon$_{\gamma=1}$ | **2.69017** | 36.91487 |
| | AdamW | 2.70994 | **36.83633** |
| | Sophia-G | 2.70653 | 41.02486 |

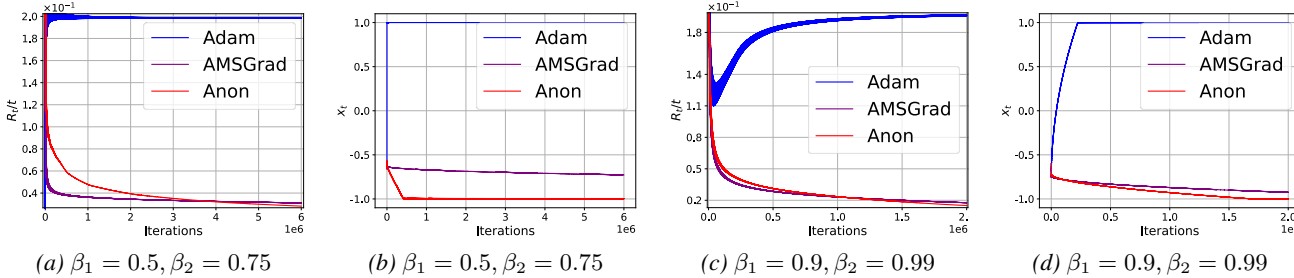

*(a)* $\beta_1 = 0.5, \beta_2 = 0.75$  *(b)* $\beta_1 = 0.5, \beta_2 = 0.75$  *(c)* $\beta_1 = 0.9, \beta_2 = 0.99$  *(d)* $\beta_1 = 0.9, \beta_2 = 0.99$

*Figure 3.* Comparison of Adam, AMSGrad, and Anon on a simple convex problem with noise. The setting of hyperparameters follows $\beta_1 < \sqrt{\beta_2}$ and $\eta(t) = 0.1/\sqrt{t}$ (Reddi et al., 2019).

**Image Generation with Diffusion Model** We conduct image generation experiments on CIFAR-10 (Krizhevsky et al., 2009) with diffusion model. We search the learning rate in $\{0.1, 0.01, 0.001, 0.0001, 0.00001\}$ for AdamW, AMSGrad, Anon, SGDM, and AdaBound. The code and the settings of other hyperparameters are consistent with the official implementation of Nichol & Dhariwal (2021). The results are reported in Table 5. When set learning rate 0.0001 (also the most suitable value for Adam) and $\gamma = 1.01$, Anon achieves SOTA and proves that the adaptivity higher than 1 is a better choice for complex models.

*Table 5.* FID scores of diffusion models on CIFAR-10 (lower is better).

| Adam | AMSGrad | SGDM | AdaBound | Anon $\gamma = 1$ | Anon $\gamma = 1.01$ |
|------|---------|------|----------|-------------------|----------------------|
| 9.11 | 8.12    | 12.84 | 12.13   | 8.03              | **7.75**             |

*Figure 4.* Hyperparameter sensitivity analysis on CIFAR-10.

**Comprehensive Analysis and Robustness** From the results on CNNs, we observe that setting the learning rate corresponding to SGDM and applying a negative adaptivity leads to better generalization and higher accuracy. In contrast, setting the learning rate equivalent to Adam and using a positive adaptivity ($\gamma \geq 1$) achieves SOTA results in diffusion models and LLMs. This observation aligns well

with our analysis in Section 2.3, highlighting that adaptivity is a key factor in model-specific optimizer behavior. Additionally, our results demonstrate the practical benefits of the proposed IDU mechanism in improving training efficiency: it accelerates computation by transforming expensive operations into negligible cost as shown in (15), and this benefit can extend to other optimizers as well. We also show the FID of setting of $\gamma = 1$ (the same as Adam) in Table 5 and Table 4 which means the only difference is the inclusion of IDU in Anon, and it also outperforms other optimizers, presenting the **improvement brought by IDU**. Furthermore, we assess the robustness of Anon to hyperparameter choices. As illustrated in Figure 4, Anon maintains high performance across a broad range of learning rates and $\gamma$ values. Notably, unlike many optimizers that require tuning of $\beta_1$, $\beta_2$, and $\epsilon$ per task, we use fixed settings for Anon ($\beta_1 = 0.9$, $\beta_2 = 0.999$, $\epsilon = 10^{-16}$) throughout all experiments. Despite this, Anon consistently achieves SOTA, validating its robustness and the practical applicability.

## 5. Conclusion

We propose Anon, a novel optimizer that obtains tunable non-restricted adaptivity and IDU convergence guarantee technique. The results of deep learning experiments show that Anon outperforms almost all other optimizers, which demonstrates the superiority of Anon and verifies the correctness of our idea about adaptivity. And we prove that Anon's convergence rate in both convex and nonconvex cases can achieve the convergence rate of mainstream optimizers under the strong assumption or with AMSGrad's technique. And the experimental results and theoretical analysis show IDU matches AMSGrad's convergence rate and memory cost. In addition, IDU offers better noise robustness, more flexibility, and even accelerates certain operations in practice. And follow the settings of those original papers, the experiments use many techniques like cosine annealing, decoupled weight decay regularization, and gradient clipping by default, so it means Anon is perfectly compatible with these widely used techniques. Thus, we expect Anon can become the preferred optimizer in extensive fields of deep learning due to its great performance.

## Impact Statement

This work significantly advances the fundamental understanding of optimization dynamics by unifying disparate adaptive methods into a continuous spectrum. By identifying and enabling previously unexplored regimes, such as "negative adaptivity" for CNNs and "super-adaptivity" for generative models, Anon unlocks new performance potentials across diverse architectures. This unified framework not only simplifies the training pipeline for the research community but also opens new theoretical avenues for analyzing loss landscapes, potentially accelerating the development of more robust and capable foundation models.

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

# Appendix

## A. Limitations and Future Work

While we have demonstrated that adaptivity is a critical attribute for first-order optimizers, our current Adaptivity Definition 2.1 covers the generic framework outlined in Algorithm 1. A small subset of first-order optimizers, such as HVAdam, deviate from this framework and are therefore not currently covered. We aim to propose a more generalized definition of adaptivity in future work to encompass these cases.

Additionally, due to computational resource constraints, our hyperparameter search for Anon was not exhaustive. For instance, in our diffusion model experiments, we observed that a learning rate of $0.0001$ combined with an adaptivity of $1.02$ hindered the reduction of training loss in the early stages. Conversely, a lower learning rate of $10^{-5}$ allowed for higher adaptivity values, such as $1.15$. Time constraints prevented a deeper investigation into these observations; thus, further research is required to fully exploit Anon's potential across different regimes.

We also hope this work contributes to the broader exploration of deep learning model design. Our experiments reveal that different model architectures exhibit distinct preferences for specific adaptivity levels. This suggests that some architectural modifications previously deemed "ineffective" might simply have been mismatched with the optimizer's fixed adaptivity (typically confined to the $[0, 1]$ range). In such scenarios, Anon's capability for extensive adaptivity tuning could potentially unlock the latent capabilities of these architectures.

## B. Adaptivity of Optimizers

### B.1. Summary of Adaptivity

We present the detailed adaptivity expressions for the optimizers discussed in the main paper in Table 6.

*Table 6.* Summary of adaptivity expressions for representative optimizers.

| Optimizer | $A_t(\psi, x)$ |
|:---:|:---:|
| SGD | $0$ |
| SGDM | $0$ |
| RMSProp | $\dfrac{1}{1 + \epsilon / \sqrt{EMA(x^2; \beta_2)}}$ |
| Adam | Same as RMSProp |
| AMSGrad | $\dfrac{1}{1 + \epsilon / \max_{i \in [t]} \sqrt{EMA(x_{1:i}^2; \beta_2)}}$ |
| Padam | $\dfrac{2p}{1 + \epsilon / \max_{i \in [t]} \sqrt{EMA(x_{1:i}^2; \beta_2)}}$ |
| AdaBound | $\begin{cases} A_t^{\text{RMSProp}}, & \text{if } \eta_l(t) < \psi_t^{\text{RMSProp}} < \eta_u(t), \\ 0, & \text{otherwise.} \end{cases}$ |
| AdaBelief | $\dfrac{1}{1 + \epsilon \cdot \frac{\frac{1}{1-\beta_2} + \sqrt{EMA((x - \phi^{\text{Adam}})^2 + \frac{\epsilon}{1-\beta_2}; \beta_2)}}{EMA((x - \phi^{\text{Adam}})^2; \beta_2)}}$ |
| Anon | Derived in (17) ($\approx \gamma$) |

For Algorithm 2, we provide an equivalent formulation in Algorithm 3. While this formulation yields no speedup, it offers a clearer representation of the underlying mechanism.

---

**Algorithm 3:** The Anon Optimizer (Equivalent Formulation)

---

1   Input: $\eta, \beta_1, \beta_2, \epsilon, \gamma$
2   **Initialize** $\boldsymbol{\theta}_0, \boldsymbol{m}_0 \leftarrow \boldsymbol{0}, \boldsymbol{s}_0 \leftarrow \boldsymbol{0}, t \leftarrow 0, k \leftarrow 0, a \leftarrow 0$
3   **while** $\boldsymbol{\theta}_t$ not converged **do**
4      $t \leftarrow t + 1$
5      $\boldsymbol{g}_t \leftarrow \nabla f_t(\boldsymbol{\theta}_t)$
6      $\boldsymbol{m}_t \leftarrow \beta_1 \boldsymbol{m}_{t-1} + (1 - \beta_1)\boldsymbol{g}_t$
7      $\widehat{\mathbf{m}}_t \leftarrow \frac{\mathbf{m}_t}{1 - \beta_2^t}$
8      $\boldsymbol{s}_t \leftarrow \beta_2 \boldsymbol{s}_{t-1} + (1 - \beta_2)\boldsymbol{g}_t^2$
9      **if** $t = 2^k$ **do**
10        $\boldsymbol{\sigma}_k \leftarrow \boldsymbol{s}_t/(1 - \beta_2^{2^k - a}) + \epsilon$
11        $a \leftarrow 2^k$
12        $\boldsymbol{v}_k \leftarrow \frac{\boldsymbol{v}_{k-1}^2 + \boldsymbol{\sigma}_k^\gamma}{2}$ if $k > 1$ else $\boldsymbol{\sigma}_k^\gamma$
13        $\boldsymbol{s}_t \leftarrow \boldsymbol{0}$
14        $\mathbf{V}_k \leftarrow \text{diag}(\sqrt{v_{k,1}}, ..., \sqrt{v_{k,d}})$
15        $k \leftarrow k + 1$
16      **end if**
17      $\boldsymbol{\theta}_t \leftarrow \Pi_{\mathcal{F}, \mathbf{V}_{k-1}}(\boldsymbol{\theta}_{t-1} - \eta(t)\mathbf{V}_{k-1}^{-1}\widehat{\boldsymbol{m}}_t)$
18 **end while**

---

## B.2. The Effect of Adaptivity

To intuitively illustrate the impact of adaptivity, we present a visualization in Figure 5.

We further demonstrate that varying adaptivity drives optimizers toward distinct regions of the parameter space by training GPT2-small on OpenWebText. The validation loss results are presented in Table 7. Additionally, we analyze the cosine similarity of the trained model parameters in Table 8. Notably, Anon with $\gamma = 1.15$ exhibits the lowest similarity when compared to Adam, Lion (Chen et al., 2023), and Muon (Jordan et al., 2024), suggesting it discovers a unique solution.

*Table 7.* Validation loss on OpenWebText.

|  | Anon$_{\gamma=1}$ | Anon$_{\gamma=1.1}$ | Anon$_{\gamma=1.15}$ | Adam | Lion | Muon |
|---|---|---|---|---|---|---|
| Loss | 2.937 | **2.927** | 2.932 | 2.934 | 2.992 | 3.092 |

## C. Experimental Details and Additional Results

### C.1. Image Classification

**ResNet18**   We report results from the sources cited in the main paper. We adopted the experimental setup from the official AdaBelief implementation[1] and reproduced the SGDM and AdaBelief results using their recommended hyperparameters. For AMSGrad (with decoupled weight decay), we searched the learning rate in {0.1, 0.01, 0.001}, finding 0.01 to be optimal. For Anon, we set the learning rate to 1 and searched $\gamma$ in {-0.1, -0.05, 0, 0.05}, with -0.1 yielding the best performance.

**ResNet50**   We report results from the sources cited in the main paper. We used the setup from the official LookAround implementation[2] and reproduced SGDM and LookAround results. Due to computational constraints, we performed a limited search for Anon, setting $\eta = 1$ and $\gamma = -0.1$.

---

[1]https://github.com/juntang-zhuang/Adabelief-Optimizer
[2]https://github.com/Ardcy/Lookaround

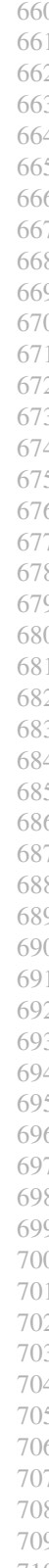

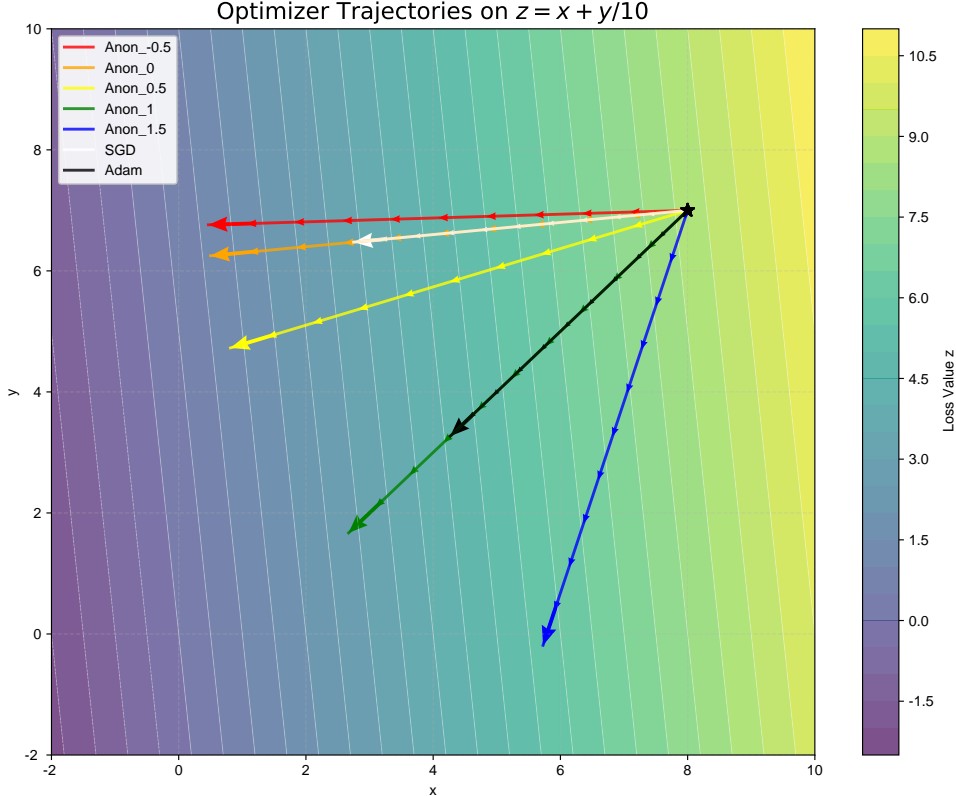

*Figure 5.* Optimization trajectories of SGDM, Adam, and Anon with varying $\gamma$. The gradient from yellow to purple indicates decreasing loss values. Different learning rates are applied to clearly visualize the distinct update directions.

*Table 8.* Cosine similarity on OpenWebText. The upper number in each cell represents the cosine similarity of **all parameters**, while the lower number represents the cosine similarity of the **weights only**.

|  | Anon$_{\gamma=1}$ | Anon$_{\gamma=1.1}$ | Anon$_{\gamma=1.15}$ | Adam | Lion | Muon |
|---|---|---|---|---|---|---|
| Anon$_{\gamma=1}$ | 1 | 0.597 | 0.355 | 0.914 | 0.857 | 0.901 |
|  | 1 | 0.317 | 0.201 | 0.511 | 0.161 | 0.194 |
| Anon$_{\gamma=1.1}$ | 0.597 | 1 | 0.425 | 0.573 | 0.522 | 0.539 |
|  | 0.317 | 1 | 0.328 | 0.248 | 0.088 | 0.098 |
| Anon$_{\gamma=1.15}$ | 0.335 | 0.425 | 1 | 0.335 | 0.299 | 0.306 |
|  | 0.201 | 0.328 | 1 | 0.156 | 0.056 | 0.063 |
| Adam | 0.914 | 0.573 | 0.355 | 1 | 0.880 | 0.923 |
|  | 0.511 | 0.248 | 0.156 | 1 | 0.191 | 0.222 |
| Lion | 0.857 | 0.522 | 0.299 | 0.880 | 1 | 0.925 |
|  | 0.161 | 0.088 | 0.056 | 0.191 | 1 | 0.132 |
| Muon | 0.901 | 0.539 | 0.306 | 0.923 | 0.925 | 1 |
|  | 0.194 | 0.098 | 0.063 | 0.222 | 0.132 | 1 |

### C.2. Image Generation

**Diffusion Model** We adopted the experimental setup from the official implementation[3] (Unconditional CIFAR-10 with L_hybrid objective and cosine noise schedule). We searched the learning rate in {0.1, 0.01, ..., 0.00001} for all optimizers. For Anon, we additionally searched $\gamma \in \{1, 1.1, 1.01\}$. The optimal configuration found was $\eta = 0.0001$ and $\gamma = 1.01$.

### C.3. Language Modeling

**GPT2** We followed the setup in the official Sophia implementation[45], setting 'nproc_per_node=4' due to resource limits. We observed that applying the Sophia learning rate scheduler to AdamW on GPT2-medium resulted in lower loss, so we applied this setting to both AdamW and Anon. We set $\gamma = 1$ for Anon. All optimizers used decoupled weight decay.

### C.4. Ablation Study on IDU Hyperparameters

We conducted an ablation study to investigate the impact of the hyperparameters $\{a_n\}$ and $\beta_3$ in IDU. Experiments were performed using ResNet20 on CIFAR-10, with results summarized in Table 9. These results demonstrate that IDU is robust to hyperparameter variations; indeed, certain configurations (e.g., $\beta_3 = 0.3, a_n = 4^{n-1}$) even outperform our default setting ($\beta_3 = 0.5, a_n = 2^{n-1}$).

*Table 9.* Ablation study on the hyperparameters $\{a_n\}$ and $\beta_3$ of IDU.

|  | $\beta_3 = 0.1$ | $\beta_3 = 0.3$ | $\beta_3 = 0.5$ | $\beta_3 = 0.7$ | $\beta_3 = 0.9$ |
|---|---|---|---|---|---|
| $a_n = 2^{n-1}$ | 91.76 | 91.98 | 92.42 | 92.43 | 92.16 |
| $a_n = 3^{n-1}$ | 92.26 | 92.23 | 92.28 | 92.34 | 92.38 |
| $a_n = 4^{n-1}$ | 91.97 | **92.44** | 92.25 | 92.12 | 92.31 |

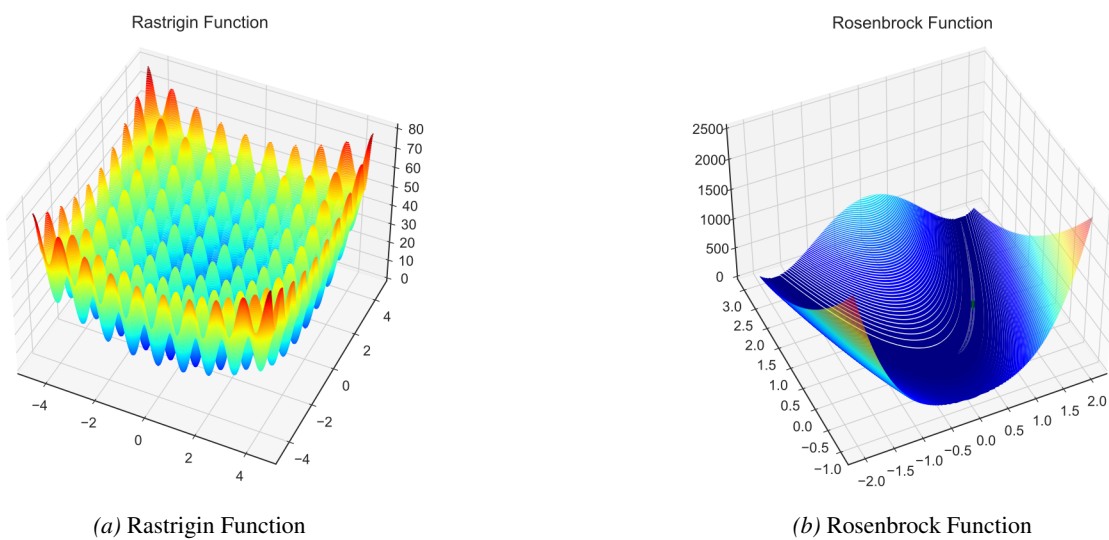

*(a)* Rastrigin Function       *(b)* Rosenbrock Function

*Figure 6.* 3D visualization of the benchmark functions.

## D. Benchmark Function Visualization

To better understand optimizer behavior in complex landscapes, we visualize their trajectories on two classical benchmark functions: Rosenbrock and Rastrigin. Rosenbrock tests the ability to follow narrow, curved valleys to a global minimum at

---

[3]https://github.com/openai/improved-diffusion

[4]https://github.com/Liuhong99/Sophia

[5]https://github.com/karpathy/nanoGPT

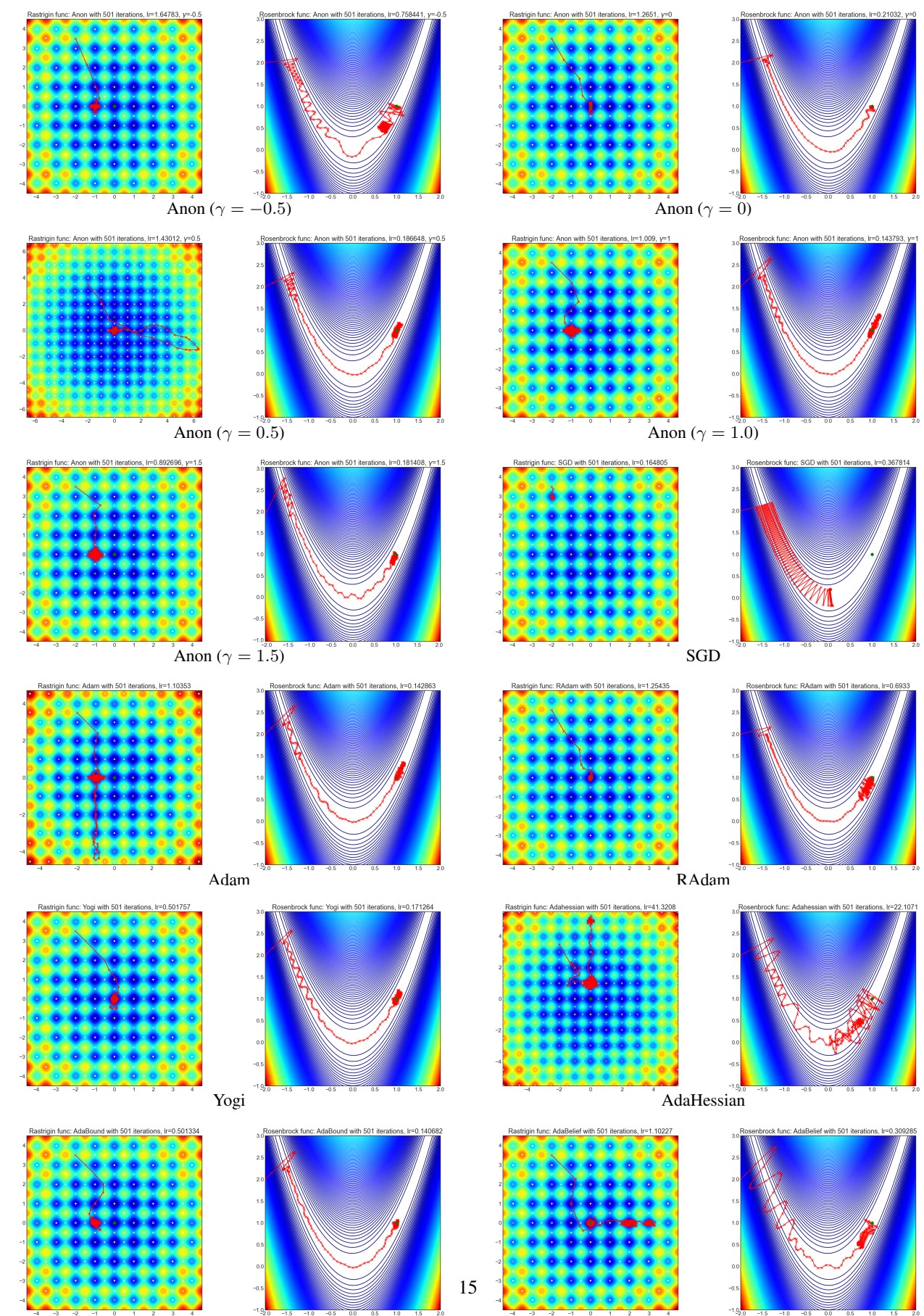

*Figure 7.* Optimization trajectory comparison using searched hyperparameters. The grid shows Rastrigin (columns 1, 3) and Rosenbrock (columns 2, 4) functions. Anon ($\gamma \leq 0$) tends to explore flatter regions, while Anon ($\gamma \geq 1$) and adaptive methods converge quickly to sharp minima.

(1, 1), evaluating stability and curvature sensitivity. Rastrigin challenges optimizers with a rugged landscape filled with deceptive local minima, with the global minimum at (0, 0).

## E. Theorem 1 in main paper

**Theorem E.1.** *If $\psi$ and $\psi'$ are from the same equivalence class, there is a function $f : \mathbb{N}_+ \to \mathbb{R}_+$ that makes $\psi_n(\boldsymbol{x}_{1:n}) = \psi'_n(\boldsymbol{x}_{1:n})f(n)$ for any $\boldsymbol{x}_{1:n} \in \mathbb{R}^n$ and any $n \in \mathbb{N}_+$.*

*Proof.* Let $h(k; \mathbf{g}_{1:n}) = \ln \psi_n(k\mathbf{g}_{1:n}) - \ln \psi'_n(k\mathbf{g}_{1:n}), h : \mathbb{R} \to \mathbb{R}$. Because $\psi_n$ and $\psi'_n$ are continuous, $h$ is continuous. When $k \neq 0$, we have

$$
\begin{aligned}
h'(k; \mathbf{x}_{1:n}) &= \lim_{\Delta k \to 0} \frac{\ln \psi_n((k + \Delta k)\mathbf{x}_{1:n}) - \ln \psi'_n((k + \Delta k)\mathbf{x}_{1:n}) - \ln \psi_n(k\mathbf{x}_{1:n}) + \ln \psi'_n(k\mathbf{x}_{1:n})}{\Delta k} \\
&= \frac{1}{k} \lim_{\Delta k \to 0} \frac{\ln \psi_n((1 + \Delta k/k)k\mathbf{x}_{1:n}) - \ln \psi'_n((1 + \Delta k/k)k\mathbf{x}_{1:n}) - \ln \psi_n(k\mathbf{x}_{1:n}) + \ln \psi'_n(k\mathbf{x}_{1:n})}{\Delta k/k} \\
&= \frac{1}{k} \lim_{\Delta k \to 0} \frac{[\ln \psi_n((1 + \Delta k/k)k\mathbf{x}_{1:n}) - \ln \psi_n(k\mathbf{x}_{1:n})] - [\ln \psi'_n((1 + \Delta k/k)k\mathbf{x}_{1:n}) - \ln \psi'_n(k\mathbf{x}_{1:n})]}{\Delta k/k} \\
&= \frac{1}{k} \Big[ A_n(\psi, \mathbf{x}_{1:n}) - A_n(\psi', \mathbf{x}_{1:n}) \Big] \\
&= \frac{1}{k} \cdot 0 \quad \Big( Since\ \psi\ and\ \psi'\ are\ in\ the\ same\ class \Big) \\
&= 0
\end{aligned} \tag{16}
$$

So $h(k; \mathbf{x}_{1:n}) = C_1$ when $k > 0$, $h(k; \mathbf{x}_{1:n}) = C_2$ when $k < 0$. And because $h$ is continuous, we have $C_1 = C_2 = h(0; \mathbf{x}_{1:n}) = \ln \frac{\psi_n(0)}{\psi'_n(0)}$.

Therefore, we have $\frac{\psi_n(k\mathbf{x}_{1:n})}{\psi'_n(k\mathbf{x}_{1:n})} = \frac{\psi_n(0)}{\psi'_n(0)}$ for $\forall k \in \mathbb{R}$.

And since $x_{0\,1:n}$ can be any vector $\in \mathbb{R}^n$ and any $n \in \mathbb{N}_+$, we have $\frac{\psi_n(\mathbf{x}_{1:n})}{\psi'_n(\mathbf{x}_{1:n})} = \frac{\psi_n(0)}{\psi'_n(0)}$ for $\forall \mathbf{x}_{1:n} \in \mathbb{R}^n$, $\forall n \in \mathbb{N}_+$.

Let $f(n) = \frac{\psi_n(0)}{\psi'_n(0)}$, we have $\psi_n(\boldsymbol{x}_{1:n}) = \psi'_n(\boldsymbol{x}_{1:n})f(n)$ for any $\boldsymbol{x}_{1:n} \in \mathbb{R}^n$ and any $n \in \mathbb{N}_+$. □

## F. Theorem 2 in main paper

**Theorem F.1.** *For the optimizer Anon described in Algorithm 2, the adaptivity of Anon in i-th dimension is $\in [\gamma(1 - k), \gamma)$, where $k = \epsilon / \min_{j \in [\tilde{a}_t]} EMA(\mathbf{g}^2_{a_{j-1}+1:a_j, i}; \beta_2)$.*

*Proof.* We let $f_{n,\gamma}(\boldsymbol{x}) = \beta_3^{-n}(1 - \beta_3 \mathbb{I}_{n>1})EMA^\gamma(\boldsymbol{x}^2_{a_{n-1}+1:a_n} + \epsilon; \beta_2)$, so we have

$$
\begin{aligned}
A(\psi, \boldsymbol{g}_{1:t,i}) &= \nabla_k \ln \left( \sum_{j=1}^{\tilde{a}_t} \beta_3^{\tilde{a}_t} f_{j,\gamma}(k\boldsymbol{g}_{1:t,i}) \right)^{1/2} \Bigg|_{k=1} \\
&= \frac{\gamma \sum_{j=1}^{\tilde{a}_t} \beta_3^{\tilde{a}_t} f_{j,\gamma-1}(\boldsymbol{g}_{1:t,i}) EMA(\boldsymbol{g}^2_{a_{j-1}+1:a_j,i}; \beta_2)}{\sum_{j=1}^{\tilde{a}_t} \beta_3^{\tilde{a}_t} f_{j,\gamma}(\boldsymbol{g}_{1:t,i})} \\
&= \frac{\gamma \sum_{j=1}^{\tilde{a}_t} \beta_3^{\tilde{a}_t} f_{j,\gamma-1}(\boldsymbol{g}_{1:t,i})[EMA(\boldsymbol{g}^2_{a_{j-1}+1:a_j,i} + \epsilon; \beta_2) - \epsilon]}{\sum_{j=1}^{\tilde{a}_t} \beta_3^{\tilde{a}_t} f_{j,\gamma}(\boldsymbol{g}_{1:t,i})} \\
&= \frac{\gamma \sum_{j=1}^{\tilde{a}_t} \beta_3^{\tilde{a}_t} f_{j,\gamma}(\boldsymbol{g}_{1:t,i}) - \gamma\epsilon \sum_{j=1}^{\tilde{a}_t} \beta_3^{\tilde{a}_t} f_{j,\gamma-1}(\boldsymbol{g}_{1:t,i})}{\sum_{j=1}^{\tilde{a}_t} \beta_3^{\tilde{a}_t} f_{j,\gamma}(\boldsymbol{g}_{1:t,i})} \\
&= \gamma \left( 1 - \epsilon \cdot \frac{\sum_{j=1}^{\tilde{a}_t} \beta_3^{\tilde{a}_t} f_{j,\gamma-1}(\boldsymbol{g}_{1:t,i})}{\sum_{j=1}^{\tilde{a}_t} \beta_3^{\tilde{a}_t} f_{j,\gamma}(\boldsymbol{g}_{1:t,i})} \right)
\end{aligned}
$$

$$= \gamma \left( 1 - \epsilon \cdot \frac{\sum_{j=1}^{\tilde{a}_t} f_{j,\gamma-1}(\boldsymbol{g}_{1:t,i})}{\sum_{j=1}^{\tilde{a}_t} f_{j,\gamma}(\boldsymbol{g}_{1:t,i})} \right) \tag{17}$$

$$\geq \gamma(1-k) \quad \left( \text{Since } k = \epsilon / \min_{j \in [\tilde{a}_t]} EMA(\mathbf{g}_{a_{j-1}+1:a_j,i}^2; \beta_2) \right) \tag{18}$$

$$\square$$

## G. Theorem 3 in main paper

For simplicity, we omit the debiasing step in theoretical analysis as in Reddi et al. (2019). It is easy to prove that the analysis also applys to the de-biased version.

**Lemma G.1.** *(McMahan & Streeter, 2010) For any $Q \in S_+^d$ and convex feasible set $\mathcal{F} \subset \mathbb{R}^d$, suppose $u_1 = \min_{x \in \mathcal{F}} \left\| Q^{1/2}(x - z_1) \right\|$ and $u_2 = \min_{x \in \mathcal{F}} \left\| Q^{1/2}(x - z_2) \right\|$, then we have $\left\| Q^{1/2}(u_1 - u_2) \right\| \leq \left\| Q^{1/2}(z_1 - z_2) \right\|$.*

**Theorem G.2.** *(Convergence analysis for online convex optimization) Let $\{\theta_t\}$ and $\{v_k\}$ be the sequence obtained by Algorithm 2, $\gamma \in \mathbb{R}$, $\beta_1 \in [0,1)$, $\beta_2 \in [0,1)$, $\beta_{1,t+1} \in [0, \beta_{1,t}]$, $\beta_{1,1} = \beta_1$, $\eta(t) = \frac{\eta_0}{\sqrt{t}}$, for $\forall t \in [T]$. Assume that $\|x - y\|_\infty \leq D_\infty$ for $\forall x, y \in \mathcal{F}$. Suppose $f(\theta)$ is a convex function, $\|g_t\|_\infty \leq G_\infty$, for $\forall t \in [T]$, $\theta \in \mathcal{F}$. Let $C_l = \min(G_\infty^{-\gamma}, \epsilon^{-\gamma})$, $C_u = \max(G_\infty^{-\gamma}, \epsilon^{-\gamma})$, where $\epsilon \in \mathbb{R}_+$ is a very number set in Algorithm 2. The optimal point of $f$ is denoted as $\theta^*$. For $\{\theta_t\}$ generated by Anon, there is a bound on the regret:*

$$\sum_{t=1}^{T} [f_t(\theta_t) - f_t(\theta^*)] \leq \frac{(1 - 2\sqrt{2})D_\infty^2}{(1 - \sqrt{2})(1 - \beta_1)C_l \eta_0} \sqrt{T} + \sum_{t=1}^{T-1} \left[ \frac{\beta_{1,t+1} \mathbb{I}_{\beta_{1,t+1} > \beta_{1,t}} D_\infty^2}{2C_l \eta_{t+1}(1 - \beta_1)^2} \right]$$

$$+ \frac{D_\infty^2}{2C_l \eta_1(1 - \beta_1)} + \frac{dD_\infty G_\infty}{1 - \beta_1} \sum_{t=1}^{T} \beta_{1,t} + \frac{dG_\infty^2 C_u \eta_0}{1 - \beta_1} \sqrt{T}$$

*Proof.*

$$\boldsymbol{v}_k = \sqrt{2/(\frac{1}{\boldsymbol{v}_{k-1}^2} + \boldsymbol{\sigma}_k^\gamma)} \text{ if } k > 0 \text{ else } \boldsymbol{\sigma}_k^{-\gamma/2}$$

$$\frac{1}{\boldsymbol{v}_k^2} = \frac{\frac{1}{\boldsymbol{v}_{k-1}^2} + \boldsymbol{\sigma}_k^\gamma}{2} \text{ if } k > 0 \text{ else } \boldsymbol{\sigma}_k^\gamma$$

$$\frac{1}{\boldsymbol{v}_k^2} = \sum_{i=0}^{k} \frac{\boldsymbol{\sigma}_i^\gamma}{2^{\min(k-i+1,k)}}$$

$$\frac{1}{\boldsymbol{v}_k^2} = \sum_{i=0}^{k} \frac{EMA^\gamma(\boldsymbol{g}_{\lfloor 2^{k-1}+1\rfloor:2^k}^2 + \epsilon; \beta_2)}{2^{\min(k-i+1,k)}} \tag{19}$$

Since $\|g_t\|_\infty \leq G_\infty$, $C_l = \min(G_\infty^{-\gamma}, \epsilon^{-\gamma})$ and $C_u = \max(G_\infty^{-\gamma}, \epsilon^{-\gamma})$, from 19, we have:

$$\frac{1}{v_{k,i}^2} \in \left[ \sum_{i=0}^{k} \frac{C_u^{-2}}{2^{\min(k-i+1,k)}}, \sum_{i=0}^{k} \frac{C_l^{-2}}{2^{\min(k-i+1,k)}} \right]$$

$$\frac{1}{v_{k,i}^2} \in \left[ C_u^{-2}, C_l^{-2} \right]$$

$$v_{k,i} \in [C_l, C_u] \tag{20}$$

Let $\eta_t = \eta(t)$.

$$\theta_{t+1} = \prod_{\mathcal{F}, V_{\tilde{a}_t}^{-1}} (\theta_t - \eta_t V_{\tilde{a}_t} m_t) = \min_{\theta \in \mathcal{F}} \left\| V_{\tilde{a}_t}^{-1/2}(\theta - (\theta_t - \eta_t V_{\tilde{a}_t} m_t)) \right\|$$

Note that $\prod_{\mathcal{F}, V_{\tilde{a}_t}^{-1}}(\theta^*) = \theta^*$ since $\theta^* \in \mathcal{F}$. Use $\theta_i^*$ and $\theta_{t,i}$ to denote the i-th dimension of $\theta^*$ and $\theta_t$ respectively. From lemma (G.1), using $u_1 = \theta_{t+1}$ and $u_2 = \theta^*$, we have:

$$
\begin{aligned}
\left\| V_{\tilde{a}_t}^{-1/2}(\theta_{t+1} - \theta^*) \right\|^2 \leq & \left\| V_{\tilde{a}_t}^{-1/2}(\theta_t - \eta_t V_{\tilde{a}_t} m_t - \theta^*) \right\|^2 \\
= & \left\| V_{\tilde{a}_t}^{-1/2}(\theta_t - \theta^*) \right\|^2 + \eta_t^2 \left\| V_{\tilde{a}_t}^{1/2} m_t \right\|^2 - 2\eta_t \langle m_t, \theta_t - \theta^* \rangle \\
= & \left\| V_{\tilde{a}_t}^{-1/2}(\theta_t - \theta^*) \right\|^2 + \eta_t^2 \left\| V_{\tilde{a}_t}^{1/2} m_t \right\|^2 \\
& - 2\eta_t \langle \beta_{1,t} m_{t-1} + (1 - \beta_{1,t}) g_t, \theta_t - \theta^* \rangle
\end{aligned}
\tag{21}
$$

Note that $\beta_1 \in [0,1)$ and $\beta_2 \in [0,1)$, rearranging inequality (21), we have:

$$
\begin{aligned}
\langle g_t, \theta_t - \theta^* \rangle \leq & \frac{1}{2\eta_t(1 - \beta_{1,t})} \left( \left\| V_{\tilde{a}_t}^{-1/2}(\theta_t - \theta^*) \right\|^2 - \left\| V_{\tilde{a}_t}^{-1/2}(\theta_{t+1} - \theta^*) \right\|^2 \right) \\
& + \frac{\eta_t}{2(1 - \beta_{1,t})} \left\| V_{\tilde{a}_t}^{1/2} m_t \right\|^2 + \frac{\beta_{1,t}}{1 - \beta_{1,t}} \langle m_{t-1}, \theta^* - \theta_t \rangle \\
\leq & \frac{1}{2\eta_t(1 - \beta_{1,t})} \left( \left\| V_{\tilde{a}_t}^{-1/2}(\theta_t - \theta^*) \right\|^2 - \left\| V_{\tilde{a}_t}^{-1/2}(\theta_{t+1} - \theta^*) \right\|^2 \right) \\
& + \frac{\eta_t}{2(1 - \beta_{1,t})} \left\| V_{\tilde{a}_t}^{1/2} m_t \right\|^2 + \frac{\beta_{1,t}}{1 - \beta_{1,t}} \left\| m_{t-1} \right\| \left\| \theta^* - \theta_t \right\| \\
& \left( \textit{Cauchy-Schwartz's inequality: } \langle u, v \rangle \leq \left\| u \right\| \left\| v \right\| \right) \\
\leq & \frac{1}{2\eta_t(1 - \beta_{1,t})} \left( \left\| V_{\tilde{a}_t}^{-1/2}(\theta_t - \theta^*) \right\|^2 - \left\| V_{\tilde{a}_t}^{-1/2}(\theta_{t+1} - \theta^*) \right\|^2 \right) \\
& + \frac{\eta_t}{2(1 - \beta_{1,t})} \left\| V_{\tilde{a}_t}^{1/2} m_t \right\|^2 + \frac{\beta_{1,t}}{1 - \beta_{1,t}} \left\| m_{t-1} \right\| \sqrt{d} D_\infty \\
& \left( \textit{Since } \left\| x - y \right\|_\infty \leq D_\infty, \textit{ for } \forall x, y \in \mathcal{F} \right) \\
= & \frac{1}{2\eta_t(1 - \beta_{1,t})} \left( \left\| V_{\tilde{a}_t}^{-1/2}(\theta_t - \theta^*) \right\|^2 - \left\| V_{\tilde{a}_t}^{-1/2}(\theta_{t+1} - \theta^*) \right\|^2 \right) \\
& + \frac{\eta_t}{2(1 - \beta_{1,t})} \left\| V_{\tilde{a}_t}^{1/2} m_t \right\|^2 + \frac{\beta_{1,t}\sqrt{d} D_\infty}{1 - \beta_{1,t}} \sqrt{\sum_{i=1}^{d} EMA^2(g_{1:t-1,i}; \beta_2)} \\
\leq & \frac{1}{2\eta_t(1 - \beta_{1,t})} \left( \left\| V_{\tilde{a}_t}^{-1/2}(\theta_t - \theta^*) \right\|^2 - \left\| V_{\tilde{a}_t}^{-1/2}(\theta_{t+1} - \theta^*) \right\|^2 \right) \\
& + \frac{\eta_t}{2(1 - \beta_{1,t})} \left\| V_{\tilde{a}_t}^{1/2} m_t \right\|^2 + \frac{\beta_{1,t}\sqrt{d} D_\infty}{1 - \beta_{1,t}} \sqrt{\sum_{i=1}^{d} G_\infty^2} \\
& \left( \textit{Since } \left\| g_t \right\|_\infty \leq G_\infty \right) \\
\leq & \frac{1}{2\eta_t(1 - \beta_{1,t})} \left( \left\| V_{\tilde{a}_t}^{-1/2}(\theta_t - \theta^*) \right\|^2 - \left\| V_{\tilde{a}_t}^{-1/2}(\theta_{t+1} - \theta^*) \right\|^2 \right) \\
& + \frac{\eta_t}{2(1 - \beta_{1,t})} \left\| V_{\tilde{a}_t}^{1/2} m_t \right\|^2 + \frac{\beta_{1,t} d D_\infty}{1 - \beta_{1,t}} G_\infty \\
= & \frac{1}{2\eta_t(1 - \beta_{1,t})} \left( \left\| V_{\tilde{a}_t}^{-1/2}(\theta_t - \theta^*) \right\|^2 - \left\| V_{\tilde{a}_t}^{-1/2}(\theta_{t+1} - \theta^*) \right\|^2 \right) \\
& + \frac{\beta_{1,t} d D_\infty G_\infty}{1 - \beta_{1,t}} + \frac{\eta_t}{2(1 - \beta_{1,t})} m_t^\top V_{\tilde{a}_t} m_t \\
= & \frac{1}{2\eta_t(1 - \beta_{1,t})} \left( \left\| V_{\tilde{a}_t}^{-1/2}(\theta_t - \theta^*) \right\|^2 - \left\| V_{\tilde{a}_t}^{-1/2}(\theta_{t+1} - \theta^*) \right\|^2 \right)
\end{aligned}
$$

$$+ \frac{\beta_{1,t} d D_\infty G_\infty}{1 - \beta_{1,t}} + \frac{\eta_t}{2(1 - \beta_{1,t})} \sum_{i=1}^{d} m_{t,i}^2 v_{\tilde{a}_t,i}$$

$$\leq \frac{1}{2\eta_t(1 - \beta_{1,t})} \left( \left\| V_{\tilde{a}_t}^{-1/2}(\theta_t - \theta^*) \right\|^2 - \left\| V_{\tilde{a}_t}^{-1/2}(\theta_{t+1} - \theta^*) \right\|^2 \right)$$

$$+ \frac{\beta_{1,t} d D_\infty G_\infty}{1 - \beta_{1,t}} + \frac{\eta_t}{2(1 - \beta_{1,t})} \sum_{i=1}^{d} m_{t,i}^2 C_u$$

$$\left( Apply \ formula \ (20) \right)$$

$$\leq \frac{1}{2\eta_t(1 - \beta_{1,t})} \left( \left\| V_{\tilde{a}_t}^{-1/2}(\theta_t - \theta^*) \right\|^2 - \left\| V_{\tilde{a}_t}^{-1/2}(\theta_{t+1} - \theta^*) \right\|^2 \right)$$

$$+ \frac{\beta_{1,t} d D_\infty G_\infty}{1 - \beta_{1,t}} + \frac{d G_\infty^2 C_u \eta_t}{2(1 - \beta_{1,t})} \tag{22}$$

By convexity of $f$, we have:

$$\sum_{t=1}^{T} f_t(\theta_t) - f_t(\theta^*) \leq \sum_{t=1}^{T} \langle g_t, \theta_t - \theta^* \rangle$$

$$\leq \sum_{t=1}^{T} \left[ \frac{1}{2\eta_t(1 - \beta_{1,t})} \left( \left\| V_{\tilde{a}_t}^{-1/2}(\theta_t - \theta^*) \right\|^2 - \left\| V_{\tilde{a}_t}^{-1/2}(\theta_{t+1} - \theta^*) \right\|^2 \right) \right.$$

$$\left. + \frac{\beta_{1,t} d D_\infty G_\infty}{1 - \beta_{1,t}} + \frac{d G_\infty^2 C_u \eta_t}{2(1 - \beta_{1,t})} \right]$$

$$\left( By \ formula \ (22) \right)$$

$$\leq \sum_{t=1}^{T} \left[ \frac{1}{2\eta_t(1 - \beta_{1,t})} \left( \left\| V_{\tilde{a}_t}^{-1/2}(\theta_t - \theta^*) \right\|^2 - \left\| V_{\tilde{a}_t}^{-1/2}(\theta_{t+1} - \theta^*) \right\|^2 \right) \right]$$

$$+ \frac{1}{1 - \beta_1} \sum_{t=1}^{T} \left( \beta_{1,t} d D_\infty G_\infty + \frac{d G_\infty^2 C_u \eta_t}{2} \right)$$

$$\left( Since \ 0 \leq \beta_{1,t} \leq \beta_1 < 1 \right)$$

$$= \sum_{t=1}^{T} \left[ \frac{1}{2\eta_t(1 - \beta_{1,t})} \left( \left\| V_{\tilde{a}_t}^{-1/2}(\theta_t - \theta^*) \right\|^2 - \left\| V_{\tilde{a}_t}^{-1/2}(\theta_{t+1} - \theta^*) \right\|^2 \right) \right]$$

$$+ \frac{1}{1 - \beta_1} \sum_{t=1}^{T} \left( \beta_{1,t} d D_\infty G_\infty + \frac{d G_\infty^2 C_u \eta_0}{2\sqrt{t}} \right)$$

$$\leq \sum_{t=1}^{T} \left[ \frac{1}{2\eta_t(1 - \beta_{1,t})} \left( \left\| V_{\tilde{a}_t}^{-1/2}(\theta_t - \theta^*) \right\|^2 - \left\| V_{\tilde{a}_t}^{-1/2}(\theta_{t+1} - \theta^*) \right\|^2 \right) \right]$$

$$+ \frac{d D_\infty G_\infty}{1 - \beta_1} \sum_{t=1}^{T} \beta_{1,t} + \frac{d G_\infty^2 C_u \eta_0}{1 - \beta_1} \int_0^T \frac{1}{2\sqrt{t}} \, \mathrm{d}t$$

$$\left( Since \ \eta_t = \eta_0/\sqrt{t} \right)$$

$$= \sum_{t=1}^{T} \left[ \frac{1}{2\eta_t(1 - \beta_{1,t})} \left( \left\| V_{\tilde{a}_t}^{-1/2}(\theta_t - \theta^*) \right\|^2 - \left\| V_{\tilde{a}_t}^{-1/2}(\theta_{t+1} - \theta^*) \right\|^2 \right) \right]$$

$$+ \frac{d D_\infty G_\infty}{1 - \beta_1} \sum_{t=1}^{T} \beta_{1,t} + \frac{d G_\infty^2 C_u \eta_0}{1 - \beta_1} \sqrt{T}$$

$$\leq \sum_{t=1}^{T-1} \left[ \frac{1}{2\eta_{t+1}(1-\beta_{1,t+1})} \left\| V_{\tilde{a}_{t+1}}^{-1/2}(\theta_{t+1}-\theta^*) \right\|^2 - \frac{1}{2\eta_t(1-\beta_{1,t})} \left\| V_{\tilde{a}_t}^{-1/2}(\theta_{t+1}-\theta^*) \right\|^2 \right]$$

$$+ \frac{1}{2\eta_1(1-\beta_1)} \left\| V_1^{-1/2}(\theta_1-\theta^*) \right\|^2 + \frac{dD_\infty G_\infty}{1-\beta_1} \sum_{t=1}^{T} \beta_{1,t} + \frac{dG_\infty^2 C_u \eta_0}{1-\beta_1} \sqrt{T}$$

$$= \sum_{t=1}^{T-1} \left[ \frac{1}{2\eta_{t+1}(1-\beta_{1,t})} \left\| V_{\tilde{a}_{t+1}}^{-1/2}(\theta_{t+1}-\theta^*) \right\|^2 - \frac{1}{2\eta_t(1-\beta_{1,t})} \left\| V_{\tilde{a}_t}^{-1/2}(\theta_{t+1}-\theta^*) \right\|^2 \right.$$

$$+ \left. \frac{\beta_{1,t+1}-\beta_{1,t}}{2\eta_{t+1}(1-\beta_{1,t})(1-\beta_{1,t+1})} \left\| V_{\tilde{a}_{t+1}}^{-1/2}(\theta_{t+1}-\theta^*) \right\|^2 \right]$$

$$+ \frac{1}{2\eta_1(1-\beta_1)} \left\| V_1^{-1/2}(\theta_1-\theta^*) \right\|^2 + \frac{dD_\infty G_\infty}{1-\beta_1} \sum_{t=1}^{T} \beta_{1,t} + \frac{dG_\infty^2 C_u \eta_0}{1-\beta_1} \sqrt{T}$$

$$= \sum_{t=1}^{T-1} \left\{ \frac{1}{2(1-\beta_{1,t})} \left[ (\theta_{t+1}-\theta^*)^\top \left( \frac{V_{\tilde{a}_{t+1}}^{-1}}{\eta_{t+1}} - \frac{V_{\tilde{a}_t}^{-1}}{\eta_t} \right) (\theta_{t+1}-\theta^*) \right] \right\}$$

$$+ \sum_{t=1}^{T-1} \left[ \frac{\beta_{1,t+1}-\beta_{1,t}}{2\eta_{t+1}(1-\beta_{1,t})(1-\beta_{1,t+1})} \left\| V_{\tilde{a}_{t+1}}^{-1/2}(\theta_{t+1}-\theta^*) \right\|^2 \right]$$

$$+ \frac{1}{2\eta_1(1-\beta_{1,t})} \left\| V_1^{-1/2}(\theta_1-\theta^*) \right\|^2 + \frac{dD_\infty G_\infty}{1-\beta_1} \sum_{t=1}^{T} \beta_{1,t} + \frac{dG_\infty^2 C_u \eta_0}{1-\beta_1} \sqrt{T}$$

$$= \sum_{k=1}^{\tilde{a}_T} \sum_{t=a_k}^{\min(T,a_{k+1})-1} \left\{ \frac{1}{2(1-\beta_{1,t})} \left[ (\theta_{t+1}-\theta^*)^\top \left( \frac{V_{\tilde{a}_{t+1}}^{-1}}{\eta_{t+1}} - \frac{V_{\tilde{a}_t}^{-1}}{\eta_t} \right) (\theta_{t+1}-\theta^*) \right] \right\}$$

$$+ \sum_{t=1}^{T-1} \left[ \frac{\beta_{1,t+1}-\beta_{1,t}}{2\eta_{t+1}(1-\beta_{1,t})(1-\beta_{1,t+1})} \left\| V_{\tilde{a}_{t+1}}^{-1/2}(\theta_{t+1}-\theta^*) \right\|^2 \right]$$

$$+ \frac{1}{2\eta_1(1-\beta_1)} \left\| V_1^{-1/2}(\theta_1-\theta^*) \right\|^2 + \frac{dD_\infty G_\infty}{1-\beta_1} \sum_{t=1}^{T} \beta_{1,t} + \frac{dG_\infty^2 C_u \eta_0}{1-\beta_1} \sqrt{T}$$

$$= \sum_{k=1}^{\tilde{a}_T} \sum_{t=a_k}^{\min(T,a_{k+1})-2} \left\{ \frac{1}{2(1-\beta_{1,t})} \left[ (\theta_{t+1}-\theta^*)^\top \left( \frac{V_{\tilde{a}_{t+1}}^{-1}}{\eta_{t+1}} - \frac{V_{\tilde{a}_t}^{-1}}{\eta_t} \right) (\theta_{t+1}-\theta^*) \right] \right\}$$

$$+ \sum_{k=1}^{\tilde{a}_T-1} \left\{ \frac{1}{2(1-\beta_{1a_{k+1}-1})} \left[ (\theta_{a_{k+1}}-\theta^*)^\top \left( \frac{V_{k+1}^{-1}}{\eta_{a_{k+1}}} - \frac{V_k^{-1}}{\eta_{a_{k+1}-1}} \right) (\theta_{a_{k+1}}-\theta^*) \right] \right\}$$

$$+ \sum_{t=1}^{T-1} \left[ \frac{\beta_{1,t+1}-\beta_{1,t}}{2\eta_{t+1}(1-\beta_{1,t})(1-\beta_{1,t+1})} \left\| V_{\tilde{a}_{t+1}}^{-1/2}(\theta_{t+1}-\theta^*) \right\|^2 \right]$$

$$+ \frac{1}{2\eta_1(1-\beta_1)} \left\| V_1^{-1/2}(\theta_1-\theta^*) \right\|^2 + \frac{dD_\infty G_\infty}{1-\beta_1} \sum_{t=1}^{T} \beta_{1,t} + \frac{dG_\infty^2 C_u \eta_0}{1-\beta_1} \sqrt{T}$$

$$= \sum_{k=1}^{\tilde{a}_T} \sum_{t=a_k}^{\min(T,a_{k+1})-2} \left\{ \frac{1}{2(1-\beta_{1,t})} \left[ (\theta_{t+1}-\theta^*)^\top \left( \frac{V_k^{-1}}{\eta_{t+1}} - \frac{V_k^{-1}}{\eta_t} \right) (\theta_{t+1}-\theta^*) \right] \right\}$$

$$+ \sum_{k=1}^{\tilde{a}_T-1} \left\{ \frac{1}{2(1-\beta_{1a_{k+1}-1})} \left[ (\theta_{a_{k+1}}-\theta^*)^\top \left( \frac{V_{k+1}^{-1}}{\eta_{a_{k+1}}} - \frac{V_k^{-1}}{\eta_{a_{k+1}-1}} \right) (\theta_{a_{k+1}}-\theta^*) \right] \right\}$$

$$+ \sum_{t=1}^{T-1} \left[ \frac{\beta_{1,t+1}-\beta_{1,t}}{2\eta_{t+1}(1-\beta_{1,t})(1-\beta_{1,t+1})} \left\| V_{\tilde{a}_{t+1}}^{-1/2}(\theta_{t+1}-\theta^*) \right\|^2 \right]$$

$$+ \frac{1}{2\eta_1(1-\beta_1)}\left\|V_1^{-1/2}(\theta_1-\theta^*)\right\|^2 + \frac{dD_\infty G_\infty}{1-\beta_1}\sum_{t=1}^{T}\beta_{1,t} + \frac{dG_\infty^2 C_u \eta_0}{1-\beta_1}\sqrt{T}$$

$$\leq \sum_{k=1}^{\tilde{a}_T}\sum_{t=a_k}^{\min(T,a_{k+1})-2}\left\{\frac{1}{2(1-\beta_1)}\left[D_\infty e_d^\top\left(\frac{V_k^{-1}}{\eta_{t+1}}-\frac{V_k^{-1}}{\eta_t}\right)D_\infty e_d\right]\right\}$$

$$+ \sum_{k=1}^{\tilde{a}_T-1}\left\{\frac{1}{2(1-\beta_1)}\left[D_\infty e_d^\top\left(\frac{C_l^{-1}\mathbf{I}_d}{\eta_{a_{k+1}}}\right)D_\infty e_d\right]\right\}$$

$$+ \sum_{t=1}^{T-1}\left[\frac{\beta_{1,t+1}-\beta_{1,t}}{2\eta_{t+1}(1-\beta_{1,t})(1-\beta_{1,t+1})}\left\|V_{\tilde{a}_{t+1}}^{-1/2}(\theta_{t+1}-\theta^*)\right\|^2\right]$$

$$+ \frac{1}{2\eta_1(1-\beta_1)}\left\|V_1^{-1/2}(\theta_1-\theta^*)\right\|^2 + \frac{dD_\infty G_\infty}{1-\beta_1}\sum_{t=1}^{T}\beta_{1,t} + \frac{dG_\infty^2 C_u \eta_0}{1-\beta_1}\sqrt{T}$$

$$\left(\textit{Since } \eta_t = \eta_0/\sqrt{t}, \textit{ and } 0 \leq \beta_{1,t} \leq \beta_1 < 1\right)$$

$$= \sum_{k=1}^{\tilde{a}_T-1}\left\{\frac{1}{2(1-\beta_1)}\left[D_\infty e_d^\top\left(\frac{V_k^{-1}}{\eta_{a_{k+1}-1}}-\frac{V_k^{-1}}{\eta_{a_k}}\right)D_\infty e_d\right]\right\}$$

$$+ \frac{1}{2(1-\beta_1)}\left[D_\infty e_d^\top\left(\frac{V_{\tilde{a}_T}^{-1}}{\eta_T}-\frac{V_{\tilde{a}_T}^{-1}}{\eta_{a_{\tilde{a}_T}}}\right)D_\infty e_d\right]$$

$$+ \sum_{k=1}^{\tilde{a}_T-1}\left\{\frac{1}{2(1-\beta_1)}\left[D_\infty e_d^\top\left(\frac{C_l^{-1}\mathbf{I}_d}{\eta_{a_{k+1}}}\right)D_\infty e_d\right]\right\}$$

$$+ \sum_{t=1}^{T-1}\left[\frac{\beta_{1,t+1}-\beta_{1,t}}{2\eta_{t+1}(1-\beta_{1,t})(1-\beta_{1,t+1})}\left\|V_{\tilde{a}_{t+1}}^{-1/2}(\theta_{t+1}-\theta^*)\right\|^2\right]$$

$$+ \frac{1}{2\eta_1(1-\beta_1)}\left\|V_1^{-1/2}(\theta_1-\theta^*)\right\|^2 + \frac{dD_\infty G_\infty}{1-\beta_1}\sum_{t=1}^{T}\beta_{1,t} + \frac{dG_\infty^2 C_u \eta_0}{1-\beta_1}\sqrt{T}$$

$$\leq 2\sum_{k=1}^{\tilde{a}_T-1}\left\{\frac{1}{2(1-\beta_1)}\left[D_\infty e_d^\top\left(\frac{C_l^{-1}\mathbf{I}_d}{\eta_{a_{k+1}}}\right)D_\infty e_d\right]\right\}$$

$$+ \frac{1}{2(1-\beta_1)}\left[D_\infty e_d^\top\left(\frac{C_l^{-1}\mathbf{I}_d}{\eta_T}\right)D_\infty e_d\right]$$

$$+ \sum_{t=1}^{T-1}\left[\frac{\beta_{1,t+1}-\beta_{1,t}}{2\eta_{t+1}(1-\beta_{1,t})(1-\beta_{1,t+1})}\left\|V_{\tilde{a}_{t+1}}^{-1/2}(\theta_{t+1}-\theta^*)\right\|^2\right]$$

$$+ \frac{1}{2\eta_1(1-\beta_1)}\left\|V_1^{-1/2}(\theta_1-\theta^*)\right\|^2 + \frac{dD_\infty G_\infty}{1-\beta_1}\sum_{t=1}^{T}\beta_{1,t} + \frac{dG_\infty^2 C_u \eta_0}{1-\beta_1}\sqrt{T}$$

$$\leq \frac{dD_\infty^2 C_l^{-1}}{(1-\beta_1)\eta_0}\sum_{k=1}^{\tilde{a}_T-1}\sqrt{a_{k+1}}$$

$$+ \frac{dD_\infty^2 C_l^{-1}}{(1-\beta_1)\eta_0}\sqrt{T}$$

$$+ \sum_{t=1}^{T-1}\left[\frac{\beta_{1,t+1}-\beta_{1,t}}{2\eta_{t+1}(1-\beta_{1,t})(1-\beta_{1,t+1})}\left\|V_{\tilde{a}_{t+1}}^{-1/2}(\theta_{t+1}-\theta^*)\right\|^2\right]$$

$$+ \frac{1}{2\eta_1(1-\beta_1)}\left\|V_1^{-1/2}(\theta_1-\theta^*)\right\|^2 + \frac{dD_\infty G_\infty}{1-\beta_1}\sum_{t=1}^{T}\beta_{1,t} + \frac{dG_\infty^2 C_u \eta_0}{1-\beta_1}\sqrt{T}$$

$$\leq \frac{dD_\infty^2 C_l^{-1}}{(1-\beta_1)\eta_0}\left(\sqrt{T}+\sum_{k=1}^{\tilde{a}_T-1}\sqrt{a_{k+1}}\right)$$

$$+\sum_{t=1}^{T-1}\left[\frac{\beta_{1,t+1}-\beta_{1,t}}{2\eta_{t+1}(1-\beta_{1,t})(1-\beta_{1,t+1})}\left\|V_{\tilde{a}_{t+1}}^{-1/2}(\theta_{t+1}-\theta^*)\right\|^2\right]$$

$$+\frac{1}{2\eta_1(1-\beta_1)}\left\|V_1^{-1/2}(\theta_1-\theta^*)\right\|^2+\frac{dD_\infty G_\infty}{1-\beta_1}\sum_{t=1}^{T}\beta_{1,t}+\frac{dG_\infty^2 C_u\eta_0}{1-\beta_1}\sqrt{T}$$

$$\leq \frac{dD_\infty^2 C_l^{-1}}{(1-\beta_1)\eta_0}\left(\sqrt{T}+\sum_{k=1}^{\tilde{a}_T-1}\sqrt{a_{k+1}}\right)+\sum_{t=1}^{T-1}\left[\frac{\beta_{1,t+1}-\beta_{1,t}}{2\eta_{t+1}(1-\beta_{1,t})(1-\beta_{1,t+1})}\left\|V_{\tilde{a}_{t+1}}^{-1/2}(\theta_{t+1}-\theta^*)\right\|^2\right]$$

$$+\frac{1}{2\eta_1(1-\beta_1)}\left(D_\infty e_d^\top V_1^{-1}D_\infty e_d\right)+\frac{dD_\infty G_\infty}{1-\beta_1}\sum_{t=1}^{T}\beta_{1,t}+\frac{dG_\infty^2 C_u\eta_0}{1-\beta_1}\sqrt{T}$$

$$\leq \frac{dD_\infty^2 C_l^{-1}}{(1-\beta_1)\eta_0}\left(\sqrt{T}+\sum_{k=1}^{\tilde{a}_T-1}\sqrt{a_{k+1}}\right)+\sum_{t=1}^{T-1}\left[\frac{\beta_{1,t+1}-\beta_{1,t}}{2\eta_{t+1}(1-\beta_{1,t})(1-\beta_{1,t+1})}\left\|V_{\tilde{a}_{t+1}}^{-1/2}(\theta_{t+1}-\theta^*)\right\|^2\right]$$

$$+\frac{dD_\infty^2 C_l^{-1}}{2\eta_1(1-\beta_1)}+\frac{dD_\infty G_\infty}{1-\beta_1}\sum_{t=1}^{T}\beta_{1,t}+\frac{dG_\infty^2 C_u\eta_0}{1-\beta_1}\sqrt{T}$$

$$\leq \frac{dD_\infty^2 C_l^{-1}}{(1-\beta_1)\eta_0}\left(\sqrt{T}+\sum_{k=1}^{\tilde{a}_T-1}\sqrt{a_{k+1}}\right)+\sum_{t=1}^{T-1}\left[\frac{\beta_{1,t+1}\mathbb{I}_{\beta_{1,t+1}>\beta_{1,t}}}{2\eta_{t+1}(1-\beta_{1,t})(1-\beta_{1,t+1})}\left\|V_{\tilde{a}_{t+1}}^{-1/2}(\theta_{t+1}-\theta^*)\right\|^2\right]$$

$$+\frac{dD_\infty^2 C_l^{-1}}{2\eta_1(1-\beta_1)}+\frac{dD_\infty G_\infty}{1-\beta_1}\sum_{t=1}^{T}\beta_{1,t}+\frac{dG_\infty^2 C_u\eta_0}{1-\beta_1}\sqrt{T}$$

$$\leq \frac{dD_\infty^2 C_l^{-1}}{(1-\beta_1)\eta_0}\left(\sqrt{T}+\sum_{k=1}^{\tilde{a}_T-1}\sqrt{a_{k+1}}\right)+\sum_{t=1}^{T-1}\left[\frac{\beta_{1,t+1}\mathbb{I}_{\beta_{1,t+1}>\beta_{1,t}}}{2\eta_{t+1}(1-\beta_1)^2}\left(D_\infty e_d^\top C_l^{-1}I_d D_\infty e_d\right)\right]$$

$$+\frac{dD_\infty^2 C_l^{-1}}{2\eta_1(1-\beta_1)}+\frac{dD_\infty G_\infty}{1-\beta_1}\sum_{t=1}^{T}\beta_{1,t}+\frac{dG_\infty^2 C_u\eta_0}{1-\beta_1}\sqrt{T}$$

$$\leq \frac{dD_\infty^2 C_l^{-1}}{(1-\beta_1)\eta_0}\left(\sqrt{T}+\sum_{k=1}^{\tilde{a}_T-1}\sqrt{a_{k+1}}\right)+\sum_{t=1}^{T-1}\left[\frac{\beta_{1,t+1}\mathbb{I}_{\beta_{1,t+1}>\beta_{1,t}}dD_\infty^2}{2C_l\eta_{t+1}(1-\beta_1)^2}\right]$$

$$+\frac{dD_\infty^2}{2C_l\eta_1(1-\beta_1)}+\frac{dD_\infty G_\infty}{1-\beta_1}\sum_{t=1}^{T}\beta_{1,t}+\frac{dG_\infty^2 C_u\eta_0}{1-\beta_1}\sqrt{T} \tag{23}$$

$$\square$$

**Corollary G.3.** *Suppose* $\beta_{1,t}=\beta_1\lambda^t,\;\;0<\lambda<1$ *in Theorem G.2, then we have:*

$$\sum_{t=1}^{T}f_t(\theta_t)-f_t(\theta^*)\leq\frac{dD_\infty^2 C_l^{-1}}{(1-\beta_1)\eta_0}\left(\sqrt{T}+\sum_{k=1}^{\tilde{a}_T-1}\sqrt{a_{k+1}}\right)+\frac{dD_\infty^2}{2C_l\eta_1(1-\beta_1)}$$

$$+\frac{dD_\infty G_\infty\beta_1}{(1-\beta_1)(1-\lambda)}+\frac{dG_\infty^2 C_u\eta_0}{1-\beta_1}\sqrt{T} \tag{24}$$

*Proof.* It is easy to prove using:

$$\sum_{t=1}^{T}\beta_{1,t}=\sum_{t=1}^{T}\beta_1\lambda^{t-1}<\sum_{t=1}^{\infty}\beta_1\lambda^{t-1}\leq\frac{\beta_1}{1-\lambda} \tag{25}$$

Plugging (25) into (23), we can derive the results above. $\square$

**Corollary G.4.** *Suppose $a_n = 2^{n-1}$, $\beta_3 = \frac{1}{2}$ in (24), then we have:*

$$\sum_{t=1}^{T} f_t(\theta_t) - f_t(\theta^*) \leq \frac{(1 - 2\sqrt{2})D_\infty^2}{(1 - \sqrt{2})(1 - \beta_1)C_l\eta_0}\sqrt{T} + \frac{D_\infty^2}{2C_l\eta_1(1 - \beta_1)}$$

$$+ \frac{dD_\infty G_\infty \beta_1}{(1 - \beta_1)(1 - \lambda)} + \frac{dG_\infty^2 C_u \eta_0}{1 - \beta_1}\sqrt{T} \tag{26}$$

*Proof.* It is easy to prove using:

$$\sum_{t=1}^{T} a^{t-1} = \frac{1 - a^T}{1 - a} \tag{27}$$

$\square$

# H. Theorem 4 in main paper

**Lemma H.1.** *(Zhuang et al., 2021) Let $m_t = \beta_1 m_{t-1} + (1 - \beta_1)g_t$, let $Q_t \in \mathbb{R}^d$, then*

$$\left\langle Q_t, g_t \right\rangle = \frac{1}{1 - \beta_1}\left(\left\langle Q_t, m_t \right\rangle - \left\langle Q_{t-1}, m_{t-1} \right\rangle\right) + \left\langle Q_{t-1}, m_{t-1} \right\rangle + \frac{\beta_1}{1 - \beta_1}\left\langle Q_{t-1} - Q_t, m_{t-1} \right\rangle \tag{28}$$

**Theorem H.2.** *(Convergence analysis for non-convex stochastic optimization) The update of $\theta_t$ can be described as $\theta_{t+1} = \theta_t - \eta_t V_{\tilde{a}_t} m_t$, and $m_t = \beta_1 m_{t-1} + (1 - \beta_1)g_t$.*
*Under the assumptions:*

- *$f$ is differentiable and $f^* \leq f \leq F$. $\nabla f(x)$ is L-Lipschitz continuous, i.e. $\|\nabla f(x) - \nabla f(y)\| \leq L\|x - y\|$, $\forall x, y$.*

- *The noisy gradient is unbias and its infinity norm is bounded by N, i.e. $\mathbb{E}g_t = \nabla f(x)$, $\|g_t\|_\infty \leq N$.*

*The hyperparameters are set as: $\eta_t = \eta_0 t^{-p}$, $\eta_0 > 0$, $p \in (0, 1)$ where the bounds are $C_l I \preceq V_{\tilde{a}_t} \preceq C_u I$, and $0 < C_l < C_u$ ($A \preceq B$ means $B - A$ is a positive semi-definite matrix). And the $\epsilon$ and $N$ ensure $C_l$ and $C_u$ exist. For sequence $\{\theta_t\}$ generated by Anon, we have:*

$$\tfrac{1}{T}\sum_{t=1}^{T}\left\|\nabla f(x_t)\right\|^2 \leq \tfrac{1}{\eta_0 C_l}T^{p-1}\left(F - f^* + K\int_1^T t^{-2p}\,\mathrm{d}t + J + K\right)$$

*where*

$$J = \frac{\beta_1^2 d}{4L(1 - \beta_1)^2}N^2 + \frac{3dN^2}{1 - \beta_1}\eta_0 C_u \sum_{k=1}^{\tilde{a}_t}\left(a_k - \mathbb{I}_{k \neq 1}\right)^{-p}, \quad K = \left(\frac{1}{1 - \beta_1} + \frac{1}{2}\right)L\eta_0^2 N^2 C_u^2 d$$

*Proof.* Let $A_t = V_{\tilde{a}_t}$, $Q_t = \eta_t A_t \nabla f(x_t)$ and let $Q_0 = Q_1$, we have

$$\sum_{t=1}^{T}\left\langle Q_t, g_t \right\rangle = \frac{1}{1 - \beta_1}\left\langle Q_T, m_T \right\rangle + \sum_{t=1}^{T}\left\langle Q_{t-1}, m_{t-1} \right\rangle + \frac{\beta_1}{1 - \beta_1}\sum_{t=1}^{T}\left\langle Q_{t-1} - Q_t, m_{t-1} \right\rangle$$

$$= \frac{\beta_1}{1 - \beta_1}\left\langle Q_T, m_T \right\rangle + \sum_{t=1}^{T}\left\langle Q_t, m_t \right\rangle + \frac{\beta_1}{1 - \beta_1}\sum_{t=0}^{T-1}\left\langle Q_t - Q_{t+1}, m_t \right\rangle \tag{29}$$

First we derive a lower bound for (29).

$$\left\langle Q_t, g_t \right\rangle = \left\langle \eta_t A_t \nabla f(x_t), g_t \right\rangle$$

$$= \left\langle \eta_{t-1} A_{t-1} \nabla f(x_t), g_t \right\rangle - \left\langle (\eta_{t-1} A_{t-1} - \eta_t A_t)\nabla f(x_t), g_t \right\rangle$$

$$\geq \left\langle \eta_{t-1} A_{t-1} \nabla f(x_t), g_t \right\rangle - \left\|\nabla f(x_t)\right\|_\infty d\left\|\eta_{t-1} A_{t-1} - \eta_t A_t\right\|_1 \left\|g_t\right\|_\infty$$

$$\left( By\ H\ddot{o}lder's\ inequality \right)$$

$$\geq \left\langle \eta_{t-1} A_{t-1} \nabla f(x_t), g_t \right\rangle - dN^2 \mathbb{I}_{t \neq a_{\tilde{a}_t}} \left( \left\| \eta_{t-1} A_{t-1} \right\| - \left\| \eta_t A_t \right\|_1 \right)$$

$$- dN^2 \mathbb{I}_{t = a_{\tilde{a}_t}} \left( \left\| \eta_{t-1} A_{t-1} - \eta_t A_t \right\|_1 \right) \tag{30}$$

$$\left( Since\ \left\| g_t \right\|_\infty \leq N,\ \eta_{t-1} \geq \eta_t > 0,\ A_{t-1} = A_t\ when\ t \neq a_{\tilde{a}_t} \right)$$

Perform telescope sum, we have

$$\sum_{t=1}^{T} \left\langle Q_t, g_t \right\rangle \geq \sum_{t=1}^{T} \left\langle \eta_{t-1} A_{t-1} \nabla f(x_t), g_t \right\rangle - dN^2 \sum_{k=1}^{\tilde{a}_T - 1} \left( \left\| \eta_{a_k} A_{a_k} \right\|_1 - \left\| \eta_{a_{k+1}-1} A_{a_{k+1}-1} \right\|_1 \right)$$

$$- dN^2 \sum_{k=1}^{\tilde{a}_T} \left\| \eta_{a_k-1} A_{a_k-1} - \eta_{a_k} A_{a_k} \right\|_1 - dN^2 \left( \left\| \eta_{a_{\tilde{a}_t}} A_{a_{\tilde{a}_t}} \right\|_1 - \left\| \eta_T A_T \right\|_1 \right)$$

$$\geq \sum_{t=1}^{T} \left\langle \eta_{t-1} A_{t-1} \nabla f(x_t), g_t \right\rangle - dN^2 \sum_{k=1}^{\tilde{a}_T - 1} \left\| \eta_{a_k} A_{a_k} \right\|_1$$

$$- dN^2 \sum_{k=1}^{\tilde{a}_T} \left\| \eta_{a_k-1} A_{a_k-1} - \eta_{a_k} A_{a_k} \right\|_1 - dN^2 \left\| \eta_{a_{\tilde{a}_t}} A_{a_{\tilde{a}_t}} \right\|_1$$

$$\geq \sum_{t=1}^{T} \left\langle \eta_{t-1} A_{t-1} \nabla f(x_t), g_t \right\rangle - dN^2 \sum_{k=1}^{\tilde{a}_T} \left\| \eta_{a_k} A_{a_k} \right\|_1$$

$$- dN^2 \sum_{k=1}^{\tilde{a}_T} \left( \left\| \eta_{a_k-1} A_{a_k-1} \right\|_1 + \left\| \eta_{a_k} A_{a_k} \right\|_1 \right)$$

$$= \sum_{t=1}^{T} \left\langle \eta_{t-1} A_{t-1} \nabla f(x_t), g_t \right\rangle - 2dN^2 \sum_{k=1}^{\tilde{a}_T} \left\| \eta_{a_k} A_{a_k} \right\|_1 - dN^2 \sum_{k=1}^{\tilde{a}_T} \left\| \eta_{a_k-1} A_{a_k-1} \right\|_1$$

$$\geq \sum_{t=1}^{T} \left\langle \eta_{t-1} A_{t-1} \nabla f(x_t), g_t \right\rangle - 3dN^2 \sum_{k=1}^{\tilde{a}_T} \eta_{a_k-1} C_u \tag{31}$$

Next, we derive an upper bound for $\sum_{t=1}^{T} \left\langle Q_t, g_t \right\rangle$ by deriving an upper-bound for the RHS of (29). We derive an upper bound for each part.

$$\left\langle Q_t, m_t \right\rangle = \left\langle \eta_t A_t \nabla f(x_t), m_t \right\rangle = \left\langle \nabla f(x_t), \eta_t A_t m_t \right\rangle$$

$$= \left\langle \nabla f(x_t), x_t - x_{t+1} \right\rangle$$

$$\leq f(x_t) - f(x_{t+1}) + \frac{L}{2} \left\| x_{t+1} - x_t \right\|^2 \tag{32}$$

$$\left( By\ L\text{-smoothness of } f \right)$$

Perform telescope sum, we have

$$\sum_{t=1}^{T} \left\langle Q_t, m_t \right\rangle \leq f(x_1) - f(x_{T+1}) + \frac{L}{2} \sum_{t=1}^{T} \left\| \eta_t A_t m_t \right\|^2 \tag{33}$$

$$\left\langle Q_t - Q_{t+1}, m_t \right\rangle = \left\langle \eta_t A_t \nabla f(x_t) - \eta_{t+1} A_{t+1} \nabla f(x_{t+1}), m_t \right\rangle$$

$$= \left\langle \eta_t A_t \nabla f(x_t) - \eta_t A_t \nabla f(x_{t+1}), m_t \right\rangle$$

$$+ \left\langle \eta_t A_t \nabla f(x_{t+1}) - \eta_{t+1} A_{t+1} \nabla f(x_{t+1}), m_t \right\rangle$$

$$= \left\langle \nabla f(x_t) - \nabla f(x_{t+1}), \eta_t A_t m_t \right\rangle + \left\langle (\eta_t A_t - \eta_{t+1} A_{t+1}) \nabla f(x_t), m_t \right\rangle$$

$$= \left\langle \nabla f(x_t) - \nabla f(x_{t+1}), x_t - x_{t+1} \right\rangle + \left\langle \nabla f(x_t), (\eta_t A_t - \eta_{t+1} A_{t+1}) m_t \right\rangle$$

$$\leq L \left\| x_{t+1} - x_t \right\|^2 + \left\langle \nabla f(x_t), (\eta_t A_t - \eta_{t+1} A_{t+1}) m_t \right\rangle$$

$$\left( \textit{By smoothness of } f \right)$$

$$\leq L \left\| x_{t+1} - x_t \right\|^2 + \left\| \nabla f(x_t) \right\|_\infty d \left\| \eta_t A_t - \eta_{t+1} A_{t+1} \right\|_1 \left\| m_t \right\|_\infty$$

$$\left( \textit{By Hölder's inequality} \right)$$

$$\leq L \left\| x_{t+1} - x_t \right\|^2 + dN^2 \mathbb{I}_{t+1 \neq a_{\tilde{a}_{t+1}}} \left( \left\| \eta_t A_t \right\|_1 - \left\| \eta_{t+1} A_{t+1} \right\|_1 \right)$$

$$+ dN^2 \mathbb{I}_{t+1 = a_{\tilde{a}_{t+1}}} \left( \left\| \eta_t A_t - \eta_{t+1} A_{t+1} \right\|_1 \right) \tag{34}$$

$$\left( \textit{Since } \eta_{t+1} \geq \eta_t > 0, A_{t+1} = A_t \textit{ when } t \neq a_{\tilde{a}_t} \right)$$

Perform telescope sum, we have

$$\sum_{t=1}^{T-1} \left\langle Q_t - Q_{t+1}, m_t \right\rangle \leq L \sum_{t=1}^{T-1} \left\| \eta_t A_t m_t \right\|^2 + dN^2 \sum_{k=1}^{\tilde{a}_T - 1} \left( \left\| \eta_{a_k} A_{a_k} \right\|_1 - \left\| \eta_{a_{k+1}-1} A_{a_{k+1}-1} \right\|_1 \right)$$

$$+ dN^2 \sum_{k=1}^{\tilde{a}_T - 1} \left\| \eta_{a_{k+1}-1} A_{a_{k+1}-1} - \eta_{a_{k+1}} A_{a_{k+1}} \right\|_1$$

$$+ dN^2 \left( \left\| \eta_{a_{\tilde{a}_T}} A_{a_{\tilde{a}_T}} \right\|_1 - \left\| \eta_T A_T \right\|_1 \right)$$

$$\leq L \sum_{t=1}^{T-1} \left\| \eta_t A_t m_t \right\|^2 + dN^2 \sum_{k=1}^{\tilde{a}_T - 1} \left\| \eta_{a_k} A_{a_k} \right\|_1$$

$$+ dN^2 \sum_{k=1}^{\tilde{a}_T - 1} \left( \left\| \eta_{a_{k+1}-1} A_{a_{k+1}-1} \right\|_1 + \left\| \eta_{a_{k+1}} A_{a_{k+1}} \right\|_1 \right)$$

$$+ dN^2 \left\| \eta_{a_{\tilde{a}_T}} A_{a_{\tilde{a}_T}} \right\|_1$$

$$\leq L \sum_{t=1}^{T-1} \left\| \eta_t A_t m_t \right\|^2 + dN^2 \sum_{k=1}^{\tilde{a}_T} \left\| \eta_{a_k} A_{a_k} \right\|_1$$

$$+ dN^2 \sum_{k=1}^{\tilde{a}_T - 1} \left( \left\| \eta_{a_{k+1}-1} A_{a_{k+1}-1} \right\|_1 + \left\| \eta_{a_{k+1}} A_{a_{k+1}} \right\|_1 \right)$$

$$\leq L \sum_{t=1}^{T-1} \left\| \eta_t A_t m_t \right\|^2 + 2dN^2 \sum_{k=1}^{\tilde{a}_T} \left\| \eta_{a_k} A_{a_k} \right\|_1$$

$$+ dN^2 \sum_{k=1}^{\tilde{a}_T - 1} \left\| \eta_{a_{k+1}-1} A_{a_{k+1}-1} \right\|_1$$

$$\leq L \sum_{t=1}^{T-1} \left\| \eta_t A_t m_t \right\|^2 + 3dN^2 \sum_{k=1}^{\tilde{a}_T} \eta_{a_k-1} C_u \tag{35}$$

We also have

$$\left\langle Q_T, m_T \right\rangle = \left\langle \eta_T A_T \nabla f(x_T), m_T \right\rangle = \left\langle \nabla f(x_T), \eta_T A_T m_T \right\rangle$$

$$\leq L\frac{1-\beta_1}{\beta_1}\left\|\eta_T A_T m_T\right\|^2 + \frac{\beta_1}{4L(1-\beta_1)}\left\|\nabla f(x_T)\right\|^2$$

$$\left(\text{By Young's inequality}\right)$$

$$\leq L\frac{1-\beta_1}{\beta_1}\left\|\eta_T A_T m_T\right\|^2 + \frac{\beta_1 d}{4L(1-\beta_1)}N^2 \tag{36}$$

Combine (33), (35) and (36) into (29), we have

$$\sum_{t=1}^{T}\left\langle Q_t, g_t\right\rangle \leq L\left\|\eta_T A_T m_T\right\|^2 + \frac{\beta_1^2 d}{4L(1-\beta_1)^2}N^2$$

$$+ f(x_1) - f(x_{T+1}) + \frac{L}{2}\sum_{t=1}^{T}\left\|\eta_t A_t m_t\right\|^2$$

$$+ \frac{\beta_1}{1-\beta_1}L\sum_{t=1}^{T-1}\left\|\eta_t A_t m_t\right\|^2 + \frac{3\beta_1}{1-\beta_1}dN^2\sum_{k=1}^{\tilde{a}_T}\eta_{a_k-1}C_u$$

$$\leq f(x_1) - f(x_{T+1}) + \left(\frac{1}{1-\beta_1}+\frac{1}{2}\right)L\sum_{t=1}^{T}\left\|\eta_t A_t m_t\right\|^2$$

$$+ \frac{\beta_1^2 d}{4L(1-\beta_1)^2}N^2 + \frac{3\beta_1}{1-\beta_1}dN^2\sum_{k=1}^{\tilde{a}_T}\eta_{a_k-1}C_u \tag{37}$$

Combine (31) and (37), we have

$$\sum_{t=1}^{T}\left\langle\eta_{t-1}A_{t-1}\nabla f(x_t), g_t\right\rangle - 3dN^2\sum_{k=1}^{\tilde{a}_t}\eta_{a_k-1}C_u \leq \sum_{t=1}^{T}\left\langle Q_t, g_t\right\rangle$$

$$\leq f(x_1) - f(x_{T+1}) + \left(\frac{1}{1-\beta_1}+\frac{1}{2}\right)L\sum_{t=1}^{T}\left\|\eta_t A_t m_t\right\|^2$$

$$+ \frac{\beta_1^2 d}{4L(1-\beta_1)^2}N^2 + \frac{3\beta_1}{1-\beta_1}dN^2\sum_{k=1}^{\tilde{a}_T}\eta_{a_k-1}C_u \tag{38}$$

Hence we have

$$\sum_{t=1}^{T}\left\langle\eta_{t-1}A_{t-1}\nabla f(x_t), g_t\right\rangle \leq f(x_1) - f(x_{T+1}) + \left(\frac{1}{1-\beta_1}+\frac{1}{2}\right)L\sum_{t=1}^{T}\left\|\eta_t A_t m_t\right\|^2$$

$$+ \frac{\beta_1^2 d}{4L(1-\beta_1)^2}N^2 + \frac{3dN^2}{1-\beta_1}\sum_{k=1}^{\tilde{a}_t}\eta_{a_k-1}C_u$$

$$\leq f(x_1) - f^* + \left(\frac{1}{1-\beta_1}+\frac{1}{2}\right)L\eta_0^2 N^2 C_u^2 d\sum_{t=1}^{T}t^{-2p}$$

$$+ \frac{\beta_1^2 d}{4L(1-\beta_1)^2}N^2 + \frac{3dN^2}{1-\beta_1}\eta_0 C_u\sum_{k=1}^{\tilde{a}_t}\left(a_k - \mathbb{I}_{k\neq1}\right)^{-p}$$

$$\leq f(x_1) - f^* + \left(\frac{1}{1-\beta_1}+\frac{1}{2}\right)L\eta_0^2 N^2 C_u^2 d\left(1 + \int_1^T t^{-2p}\,\mathrm{d}t\right)$$

$$+ \frac{\beta_1^2 d}{4L(1-\beta_1)^2}N^2 + \frac{3dN^2}{1-\beta_1}\eta_0 C_u\sum_{k=1}^{\tilde{a}_t}\left(a_k - \mathbb{I}_{k\neq1}\right)^{-p}$$

$$\leq f(x_1) - f^* + \left(\frac{1}{1-\beta_1}+\frac{1}{2}\right)L\eta_0^2 N^2 C_u^2 d\int_1^T t^{-2p}\,\mathrm{d}t$$

$$+ \underbrace{\frac{\beta_1^2 d}{4L(1-\beta_1)^2}N^2 + \frac{3dN^2}{1-\beta_1}\eta_0 C_u \sum_{k=1}^{\tilde{a}_t}(a_k - \mathbb{I}_{k\neq 1})^{-p}}_{J}$$

$$+ \underbrace{\left(\frac{1}{1-\beta_1} + \frac{1}{2}\right)L\eta_0^2 N^2 C_u^2 d}_{K}$$

$$\leq f(x_1) - f^* + K\int_1^T t^{-2p}\,\mathrm{d}t + J + K \tag{39}$$

Take expectations on both sides, we have

$$\sum_{t=1}^T \left\langle \eta_{t-1}A_{t-1}\nabla f(x_t), \nabla f(x_t)\right\rangle \leq \mathbb{E}f(x_1) - f^* + K\int_1^T t^{-2p}\,\mathrm{d}t + J + K$$

$$\leq F - f^* + K\int_1^T t^{-2p}\,\mathrm{d}t + J + K \tag{40}$$

Note that we have $\eta_t$ decays monotonically with $t$, hence

$$\sum_{t=1}^T \left\langle \eta_{t-1}A_{t-1}\nabla f(x_t), \nabla f(x_t)\right\rangle \geq \eta_0 T^{-p}\sum_{t=1}^T \left\langle A_{t-1}\nabla f(x_t), \nabla f(x_t)\right\rangle \tag{41}$$

$$\geq \eta_0 T^{1-p}C_l \frac{1}{T}\sum_{t=1}^T \left\|\nabla f(x_t)\right\|^2 \tag{42}$$

Combine (40) and (42), assume $f$ is upper bounded by $M_f$, we have

$$\frac{1}{T}\sum_{t=1}^T \left\|\nabla f(x_t)\right\|^2 \leq \frac{1}{\eta_0 C_l}T^{p-1}\left(F - f^* + K\int_1^T t^{-2p}\,\mathrm{d}t + J + K\right) \tag{43}$$

And it is easy to proved when $a_n = 2^{n-1}$, we have

$$J = \frac{\beta_1^2 d}{4L(1-\beta_1)^2}N^2 + \frac{3dN^2}{1-\beta_1}\eta_0 C_u \sum_{k=1}^{\tilde{a}_t}(a_k - \mathbb{I}_{k\neq 1})^{-p} \tag{44}$$

$$\leq \frac{\beta_1^2 d}{4L(1-\beta_1)^2}N^2 + \frac{3dN^2}{1-\beta_1}\eta_0 C_u \sum_{k=1}^{\infty}(a_k - \mathbb{I}_{k\neq 1})^{-p} \tag{45}$$

$$= \frac{\beta_1^2 d}{4L(1-\beta_1)^2}N^2 + \frac{3dN^2}{1-\beta_1}\eta_0 C_u \left(1 + \sum_{k=2}^{\infty}(2^{k-1} - \mathbb{I}_{k\neq 1})^{-p}\right) \tag{46}$$

$$\leq \frac{\beta_1^2 d}{4L(1-\beta_1)^2}N^2 + \frac{3dN^2}{1-\beta_1}\eta_0 C_u \left(1 + \sum_{k=2}^{\infty}(2^{k-2})^{-p}\right) \tag{47}$$

$$= \frac{\beta_1^2 d}{4L(1-\beta_1)^2}N^2 + \frac{3dN^2}{1-\beta_1}\eta_0 C_u \left(1 + \sum_{k=1}^{\infty}(2^{-p})^{k-1}\right) \tag{48}$$

$$= \frac{\beta_1^2 d}{4L(1-\beta_1)^2}dN^2 + \frac{3dN^2}{1-\beta_1}\eta_0 C_u \left(1 + \frac{1}{1-2^{-p}}\right) \tag{49}$$

$\square$

