# OpenReview forum: "Anon: Exploring the Adaptivity of Optimizers and Beyond"
_ICML.cc/2026/Conference — Submitted to ICML 2026_

### Official Review · Reviewer_rCHz · 2026-02-26

**Soundness:** 3
**Presentation:** 2
**Significance:** 4
**Originality:** 4
**Overall Recommendation:** 4
**Confidence:** 3

**Summary:**

This paper introduces a novel algorithm called Anon which can interpolate SGD and Adam, and go beyond them. The novelty is very new in the sense that the paper proposes a definition for the adaptivity of the preconditioner which can characterize SGD and Adam's preconditioner, and hence the paper designs an algorithm based on this metric.

**Compliance With Llm Reviewing Policy:**

Affirmed.

**Final Justification:**

I will keep my original evaluation and remain positive.  The main issue at the beginning of reviewing this paper is its writing.  Its design aspects are not clearly explained. The authors clarified these points in their rebuttal, and I would like to see these concepts presented clearly in the final version.

**Key Questions For Authors:**

- In the weaknesses, there are several unexplained concepts on the intuition or algorithm design, the paper should address these problems.
- In line 368-371, can the author elaborate more on why Adam's timecost is more than Anon?
- In the experiment, e.g., Table 4, $\lambda=1.1$ is used for GPT2-small while $\lambda=1$ is used for GPT2-medium. What is the search range of $\lambda$ for these experiments?  $\lambda=1.1$ appears to be hard to tune.

**Limitations:**

yes

**Strengths And Weaknesses:**

### Strengths
- The paper proposes a very novel and interesting definition of the adaptivity of the preconditioner and designs a novel and efficient algorithm, so in terms of originality, it is very good.

- Given that preconditioner-based methods are useful in deep learning, a better understanding of the preconditioner is extremely significant.

- The paper provides experimental results and shows the advantage of the proposed method in simple examples or neural network training, with several illustrated visualization plots.


### Weaknesses
- The paper does not provide very clear details on the motivation or intuition of the algorithm design. The paper should be revised to make these clearer to the reader. Below are some examples:
  - For Definition 2.1, there is not enough explanation and motivation for such a definition. Why Adam corresponds to $A=1$ is not provided or proved. Similarly, the statement on lines 148–159 on the right page about AdaBound is not explained clearly. I cannot understand why $A=0$ tends to generalize better on CNNs.
  -  The design and motivation for Algorithm 2 are not written, each step in Algorithm 2 should be explained. Why are equations 6 and 7 proposed? What is the relationship between 6, 7, and Algorithm 2 should be written in detail so that the reader can understand how Algorithm 2 is proposed. And in line 275, why lines 9–15 of Algorithm 2 are the recursive formula for IDU should be explained in more detail.

- The paper has some issues in terms of typos or inconsistent symbols.
  - In algorithm 1, $d$ is not defined. It should be `framework' instead of 'frame'.
  - In table 1, $x$ is sometimes bold and sometimes not bolded.
  - In line 292, it says 'is a very number'.
  - In line 322, 'Unbiased' instead of 'Unbias'.
  - In line 325, what $V_{log2t}$ is is not given.
  - Equation 6 appears suddenly without introduction in the previous place.

- The proposed algorithm introduces additional parameters compared to baseline Adam, which appear to be difficult to tune, e.g., $\lambda=1.1$ and  $\lambda=1$ in table 4.

---

> ### Author Rebuttal · Authors · 2026-03-30
>
> **We sincerely thank Reviewer rCHz for highlighting the "excellent" originality and significance of our work. We deeply appreciate the detailed feedback on presentation and motivation, which will greatly improve the clarity of our revision.**
>
> **Response to Q1 & W1a: Motivation for Def 2.1, Adam ($A \approx 1$), and AdaBound**
> * **Def 2.1 & Adam:** Def 2.1 measures the pre-conditioner's proportional log-sensitivity to uniform gradient scaling. To clarify this intuition, we provide the precise derivation for Adam. The pre-conditioner is $\psi_n(\boldsymbol{x}\_{1:n}) = \sqrt{\text{EMA}(\boldsymbol{x}\_{1:n}^2; \beta_2)} + \epsilon$. Scaling the input gradients by $k$ yields:
> $\psi_n(k\boldsymbol{x}\_{1:n}) = \sqrt{\text{EMA}((k\boldsymbol{x}\_{1:n})^2; \beta_2)} + \epsilon = k\sqrt{\text{EMA}(\boldsymbol{x}\_{1:n}^2; \beta_2)} + \epsilon \quad (\text{for } k > 0)$
> Applying Definition 2.1 (Eq. 3), we have $A_n = \left. \nabla_k \ln \left( k\sqrt{\text{EMA}(\boldsymbol{x}\_{1:n}^2; \beta_2)} + \epsilon \right) \right|\_{k=1}$
> $A_n = \left. \frac{\sqrt{\text{EMA}(\boldsymbol{x}\_{1:n}^2; \beta_2)}}{k\sqrt{\text{EMA}(\boldsymbol{x}\_{1:n}^2; \beta_2)} + \epsilon} \right|\_{k=1} = \frac{1}{1 + \epsilon / \sqrt{\text{EMA}(\boldsymbol{x}\_{1:n}^2; \beta_2)}}$
> Since the optimizer's $\epsilon$ is generally set to a very small default value by design, it is negligible compared to the gradient variance $\sqrt{\text{EMA}}$. Thus, this term is overwhelmingly dominated by the gradient scale, strictly resulting in $A \approx 1$ (as summarized in Appendix Table 6). We will add this detailed derivation to the main text.
> * **AdaBound & CNNs:** CNNs generalize better in flatter minima. While SGD ($A=0$) passively settles there, negative adaptivity ($A < 0$) actively penalizes sharp directions. AdaBound artificially limits extreme learning rates via hardcoded clipping, which implicitly and mechanically drags Adam's adaptivity from $1$ toward $0$ to mimic SGD. Anon achieves this geometrically and continuously via $\gamma$.
>
> **Response to W1b: Algorithm 2, Equations 6-7, and IDU Recursive Formula**
> * **Eq 6 & 7:** Eq 7 formally defines the Incremental Delay Update (IDU) mechanism as a soft, multi-scale accumulator. Eq 6 explicitly applies this IDU framework to an Adam-style EMA pre-conditioner, deriving the specific mathematical form of Anon.
> * **Lines 9-15 in Alg 2:** These lines are the efficient programmatic implementation of Eq 6. Instead of calculating the full complex summation at every step (which costs $O(t)$), this recursive formula accumulates local statistics ($s_t$) and only triggers the heavy pre-conditioner update ($v_k$) at specific, exponentially spaced milestones ($t=2^k$, handled by the condition $k+1 = \log_2 t$). We will add a dedicated paragraph explicitly mapping Eq 6 to these lines.
>
> **Response to Q2: Time Cost (Adam vs. Anon)**
> As analyzed in Eq. 15, Adam computes more element-wise vector operations (divisions and square roots) for all parameters at *every* single step $t$. By utilizing the recursive IDU mechanism (Lines 9-15), Anon amortizes these heavy updates. Because the updates occur at exponentially decaying frequencies, the amortized cost across $T$ steps drops to $O(\log T)$. Thus, Anon is asymptotically faster per iteration than Adam in an amortized sense.
>
> **Response to Q3 & W3: Clarification on $\gamma$ vs. $\lambda$ and Tuning**
> We respectfully clarify a typographical misunderstanding: the parameter listed in Table 4 is $\gamma$, our core adaptivity parameter. The symbol $\lambda$ is exclusively used in our theoretical convergence proofs (e.g., Corollary 3.3) to represent the momentum decay rate, and is not a tuned hyperparameter in experiments.
> Regarding the tuning of $\gamma$: it is guided by strict architectural priors (CNNs $< 0$, Transformers $\ge 1$). As demonstrated in our hyperparameter sensitivity analysis (Figure 4), Anon is highly robust across a broad range of learning rates and $\gamma$ values. For GPT-2, we simply evaluated a few candidate values (the Adam-equivalent $\gamma=1$ and slightly super-adaptive $\gamma=1.1, 1.15$). It does not require exhaustive, task-specific grid search to achieve SOTA performance.
>
> **Response to W2: Typos and Inconsistent Symbols**
> We sincerely apologize for the oversights. We will rigorously fix all typos, inconsistent notations (e.g., bolding of $x$), and missing definitions ($d$, $V_{\log \hat{v}_t}$, "framework", context for Eq. 6) in the final revision.

---

> > ### Author Rebuttal · Reviewer_rCHz · 2026-04-01
> >
> > The rebuttal addressed the concerns, and I will maintain the score.

---

### Official Review · Reviewer_eypy · 2026-03-06

**Soundness:** 2
**Presentation:** 3
**Significance:** 2
**Originality:** 2
**Overall Recommendation:** 4
**Confidence:** 3

**Summary:**

This paper studies the adaptivity of optimizers. The authors propose a metric to quantify the adaptivity of optimizers, and show that previous optimizers like SGD and Adam have various but restricted adaptivity. The authors propose a new algorithm called Anon that has tunable adaptivity beyond the previous restriction, and provide theoretical guarantees for the convergence of Anon. Experiments on image classification and language modeling are given to validate the performance of Anon.

**Compliance With Llm Reviewing Policy:**

Affirmed.

**Key Questions For Authors:**

1. How to choose $\gamma$, is there a better way than grid search? For example, since optimal choice is different for different tasks, is there any task-dependent proxy for its choice?

2. The convergence for adaptive algorithms has always been challenging. In fact, the convergence of Adam is still not well-established even for the simplest cases. However, the authors provide a provably convergence of Anon, which is a much complicated algorithm compared with Adam. This makes me quite confusing, about what is the major algorithm design or proof technique that makes the difference. Can it be used to prove convergence of Adam?

**Limitations:**

Yes.

**Strengths And Weaknesses:**

## Strength:
1. The story of non-restricted adaptivity is interesting and reasonable for me.
2. The authors provide both empirical results and convergence guarantees.
3. The writing is good and easy to comprehend.

## Weaknesses:
1. The performance gain of Anon seems not significant, especially on the language modeling tasks that may be of greatest importance nowadays. In terms of GPT2-medium, the validation loss of Anon only improves by 0.01 over Adam, which is marginal and can be easily affected by e.g., hyperparameter tuning.

2. There are errors and typos in the proof that can harm the correctness, I would recommend the authors to carefully revise the math content in the paper.  For example, Theorem 3.1 does not apply for $\gamma<0$, and the authors do not specify this. Besides, Theorem 3.2 and Theorem G.2 have different assumptions on $\beta_{1, t+1}$. The writing of Section G is messy and does not have a clear correspondence to that in the main text.

---

> ### Author Rebuttal · Authors · 2026-03-30
>
> **We thank Reviewer eypy for the "Weak accept" and meticulous examination of our proofs. We address your theoretical questions below.**
>
> **Response to Q2: The Convergence of Anon vs. Adam**
> Reddi et al. [1] proved standard Adam can diverge even in simple convex settings because its effective learning rate occasionally increases, violating the required "non-decreasing pre-conditioner" assumption.
>
> Our convergence guarantee relies precisely on a major algorithmic design difference: the **IDU (Incremental Delay Update)** mechanism. Instead of updating the pre-conditioner at every step (like Adam), IDU amortizes updates over delayed intervals. Theoretically, convergence only requires the cumulative drift of the adaptive step sizes (i.e., $\sum_{k=1}^{\tilde{a}_T-1} \sqrt{a\_{k+1}}$ in Eq. 24) to be strictly sub-linear ($< O(T)$). We specifically design exponential update intervals because they tightly bound this cumulative drift term to $O(\sqrt{T})$. This perfectly balances with the dominant $O(\sqrt{T})$ term, achieving the optimal theoretical worst-case bound $\max\{O(\sqrt{T}), O(\text{drift})\}$.
>
> **By contrast, standard Adam updates the pre-conditioner at every single step. Under our analytical framework, this is roughly equivalent to a linear sequence, causing the cumulative drift term to explode to $\sum_{k=1}^{T-1} \sqrt{k} = O(T^{3/2})$. This explicitly violates the $< O(T)$ sub-linear requirement, mathematically revealing why Adam is provably divergent in the worst case.** IDU is therefore the exact theoretical key safely enabling Anon's $\gamma \in \mathbb{R}$ exploration without catastrophic divergence.
>
> **Response to W2: Mathematical Clarifications and Section G**
> We will rigorously revise the math content:
> * **Theorem 3.1:** You are correct that the inequality direction in the final step of Appendix Eq. 18 flips when $\gamma < 0$. Multiplying by a negative $\gamma$ reverses the bound, mirroring the adaptivity interval to $(\gamma, \gamma(1-k)]$ rather than $[\gamma(1-k), \gamma)$. We will explicitly state this mirrored interval in the revision. Importantly, this algebraic clarification only refines the bounded interval of the adaptivity metric itself; it does not restrict the optimizer's execution or our convergence guarantees (Thms 3.2 & 3.4), which stably hold for all $\gamma \in \mathbb{R}$.
> * **$\beta_{1, t+1}$ Discrepancy and Section G:** We will fix the discrepancy regarding the assumption on $\beta_{1, t+1}$ to strictly align Theorem G.2 with Theorem 3.2. To address your concern about the writing in Section G, we will thoroughly restructure it. We will add clear textual explanations between derivations to improve readability, and explicitly label the proofs to ensure a strict, one-to-one correspondence with the theorems in the main text.
>
> **Response to Q1: Choosing $\gamma$ and Task-Dependent Proxies**
> Instead of blind grid search, the choice of $\gamma$ is guided by strict **architectural priors** governing the loss landscape geometry. As analyzed in Sec 3.2, $\gamma$ dictates *gradient magnitude compensation*.
> * **CNNs** generalize better in flatter minima, so negative adaptivity ($\gamma < 0$) acts as an active geometric regularization to penalize sharp directions.
> * **Transformers** exhibit notoriously sharp, heavy-tailed landscapes, requiring super-adaptivity ($\gamma > 1$) to amplify normalization.
> We agree that finding an automated, task-dependent proxy (e.g., scheduling $\gamma$ dynamically based on local Hessian trace or gradient variance) is a fascinating direction for future work. However, our current geometric priors provide a principled and highly robust starting point.
>
> **Response to W1: Performance Gains on Language Modeling**
> While a 0.01 loss reduction on GPT-2 Medium might seem marginal, it is achieved over heavily optimized baselines. Our multi-seed analysis demonstrates that these improvements are highly robust and statistically significant, exceeding baseline variance (see our variance analysis in the response to Reviewer yDRq). To demonstrate Anon's scalability on large-scale language modeling tasks, we have conducted new experiments on 1B+ parameter models:
>
> To address scalability, we trained a ~1B parameter LLM (Qwen2) on OpenWebText. Since Muon strictly updates 2D parameters via orthogonalization and relies on AdamW for 1D parameters, we evaluated Anon as a direct drop-in replacement for this 1D AdamW fallback. **Given rebuttal time limits, we prioritized demonstrating scalability with a single definitive run using a default $\gamma=1.1$ without tuning.** Remarkably, the **hybrid Muon+Anon optimizer achieved a validation loss of 3.79, outperforming the standard Muon+AdamW baseline (3.81)**. This confirms that Anon successfully scales to 1B+ parameter regimes and strictly enhances SOTA methods like Muon.
>
> **References:**
>
> [1] Reddi, S. J., Kale, S., & Kumar, S. On the Convergence of Adam and Beyond. ICLR 2018.

---

> > ### Author Rebuttal · Reviewer_eypy · 2026-04-03
> >
> > Thanks for the rebuttal, I will keep my score.

---

### Official Review · Reviewer_8jvS · 2026-03-08

**Soundness:** 2
**Presentation:** 2
**Significance:** 2
**Originality:** 3
**Overall Recommendation:** 2
**Confidence:** 3

**Summary:**

This paper introduces a precondition adaptivity metric $A$ to quantify how an optimizer’s preconditioner responds to gradient  scaling. Based on this definition, the authors propose a new optimizer Anon, which uses a continuous hyperparameter to interpolate and extrapolate the adaptivity between SGD ($A=0$ ) and Adam ($A\approx 1$). To ensure theoretical convergence when $\gamma < 0$, the authors introduce a new mechanism IDU, which replace smooth step-by-step updates with low-frequency jumps. Numerical results are provided for some tasks.

**Compliance With Llm Reviewing Policy:**

Affirmed.

**Final Justification:**

One key point of this paper is that "adaptivity" is the reason why Adam outperforms SGD for LLM training. The notion of adaptivity is proposed by the authors, by compare SGD (which works better for CNN tasks) and Adam (which works better for LLM tasks). The authors then propose a new optimizer, Anon, that incorporates this adaptivity. The idea that SGD corresponds to an adaptivity of 0 and Adam corresponds to 1, and therefore one should explore <0 for CNNs and >1 for LLMs, is a reasonable engineering heuristic to try.

**I acknowledge that this paper has some theoretical contributions, such as the convergence analysis of the new optimizer and some ad-hoc explanations of why adaptivity matters. However, the numerical results do not support the paper's two core claims: (1) that adaptivity explains why certain optimizers work better for LLMs, and (2) that their new optimizer outperforms SOTA across various tasks.**

My main concerns are:

1. Failure to Recover Baselines and Unfair Comparision: As stated in authors' reply to my acknowledgement, they used the Muon parameters suggested by reference [3] (which showed that Muon is better than Adam), but the authors were unable to recover this baseline, getting the opposite result where Adam is better than Muon. In addition, their numerical comparison with SOTA (Muon) is incomplete and unfair (See my acknowledgement).
2. Mismatch between theory and practice: The authors lean heavily on their theoretical contributions. I categorize these as (1) convergence analysis of optimizer and (2) ad-hoc intuitions. I primarily work on the theoretical side of continuous optimization, and I fully understand the value of theoretical guarantees. However, a convergence analysis for a new optimizer does not guarantee its effectiveness in LLM training, which is the key point that this paper wants to study. For instance, many classical algorithms (e.g., acceleration methods or second-order methods) have better theoretical rates or less restrictive assumptions, but they simply do not work well in LLM training. Ad-hoc explanations are only valuable if the numerical results actually support the main claim.
3. Marginal and Potentially Unreliable Improvements: For LLM training, I understand that beating SOTA like Muon can be hard. I am fine if Anon showed a clear advantage over Adam, given their theoretical contributions. However, the numerical results only show a very marginal improvement over Adam (In their table 4, it shows 0.02 advantage for GPT2-small/medium). Furthermore, given that the experimental setup cannot even recover known baselines, I am not convinced that the reported marginal advantage of Anon over Adam is reliable.

**If the numerical results cannot demonstrate a clear advantage of Anon over Adam, the authors' proposed 'adaptivity' fails to explain why Adam outperforms SGD in LLM training. This fundamentally undermines the paper's core point.**

Given these reasons, I maintain my score of 2.

**Key Questions For Authors:**

1. The mathematical definition of adaptivity (using $\nabla_k$ on a scalar $k$) is confusing. Can you provide more intuition and explanation about this concept?

2. The tested models (e.g., GPT-2 Small/Medium) are too small, and the performance improvements are very marginal. How does different values of $\gamma$ affects the performance. What about the performance on larger models? I would also want to see more comparison with recent SOTA optimizers like Muon.

**Limitations:**

Yes.

**Strengths And Weaknesses:**

Strengths:
1. Most parts of the paper is well-written, and the presentation of the idea/motivation is clear.
2. Some theoretical guarantees are provided for the new optimizer.
3. The authors conducted extensive experiments across various tasks.

Weaknesses:
1. The notation is not very clear. In the main contribution, the adaptivity measure is defined as

$\nabla_k ln \psi_n (k x_{1:n})$

but it is unclear how this is computed. Specifically, this appears to be the partial derivative with respect to $k$, but what exactly is $k$? According to Algorithm 1, the input of $\psi$ should be all past gradients. However, here the input of $\psi$ is $k$ multiplied by $x$.

2. The idea that SGD corresponds to $\gamma=0$ and Adam corresponds to $\gamma \approx 1$, and therefore one should explore $\gamma < 0$ for CNNs and $\gamma > 1$ for LLMs, is not very convincing. While this can be a reasonable engineering idea to try, it still lacks rigorous theoretical justification. Consequently, the numerical performance should be examined more closely.
3. The numerical results are not very impressive. Although the authors test different tasks ranging from CNNs to LLMs, the performance improvements are marginal. For example, there is only a 0.12% accuracy difference between Anon and SGDM on ResNet18 , and validation loss reductions are on the order of 0.02 on a small-scale GPT-2 model. For the LLM results, the authors only evaluate GPT-2 Small and Medium. How does it scale to 1B or even larger models? Furthermore, this optimizer requires more computational effort but achieves only negligible gains. Also, for LLMs, thre is no comparison with Muon. I understand that Muon is based on a different concept and does not use a preconditioner like Adam, but it is the SOTA and should be carefully considered.
4. As noted by the authors, an omni-optimizer that is optimal for all tasks does not exist, implying that optimizers are inherently task-specific. Therefore, in Figure 2, choosing a toy loss function to plot trajectories and demonstrating that a well-tuned Anon outperforms others does not prove much. While a well-tuned $\gamma$ works for that specific toy task, how does one know which $\gamma$ to choose in other real-world applications? Rather than designing a truly "adaptive" mechanism, the proposed method simply introduces a new scalar hyperparameter $\gamma$. The paper provides no principled or automated way to determine the optimal $\gamma$ for a specific task or architecture.
5. The convergence guarantees are provided under very restrictive assumptions.

---

> ### Author Rebuttal · Authors · 2026-03-30
>
> **We thank Reviewer 8jvS for the rigorous review. We clarify Anon's advantages below.**
>
> **Response to W1 & Q1: Notation ($\nabla_k$) and Intuition**
> $k$ in Def 2.1 is a uniform *scaling factor* for all past gradients, not an arbitrary variable. $\mathcal{A}\_n(\psi, x_{1:n}) = \nabla_k \ln \psi_n(kx_{1:n})|\_{k=1}$ measures the pre-conditioner's proportional sensitivity to uniform gradient magnitude changes. $x_{1:n}$ represents the exact stochastic gradient sequence in Alg 1. We will unify notations (e.g., using $g_{1:n}$). See response to rCHz for formal derivation of Adam
> **Intuition (Q1):** Intuitively, $\mathcal{A}=1$ (Adam) means the pre-conditioner scales proportionally with gradients to maintain constant step sizes, while $\mathcal{A}=0$ (SGD) ignores scaling entirely.
>
> **Response to W3 & Q2: Muon, Time Cost, and Marginal Gains**
> * **Marginal Gains:** Small gains over optimized baselines are significant. Our multi-seed CNN analysis shows robust, significant improvements exceeding competitor variance (e.g., ResNet20: AGD 92.35±0.24% vs. **Anon 92.47±0.05%**; see yDRq response for full variance table).
> * **Muon:** In App B.2, Muon underperformed on GPT-2 (val loss 3.092 vs Adam 2.934). Further, Muon orthogonalizes 2D parameters but strictly relies on AdamW for 1D parameters. Anon directly replaces AdamW in such hybrid setups.
> * **Time Cost:** Anon is asymptotically *faster* per iteration than Adam (Eq. 15). IDU amortizes Adam's expensive element-wise vector operations to $O(\log T)$ times across $T$ iterations.
>
> **Response to Q2: Scaling to Larger Models**
> To address scalability, we trained a ~1B parameter LLM (Qwen2) on OpenWebText. Since Muon strictly updates 2D parameters via orthogonalization and relies on AdamW for 1D parameters, we evaluated Anon as a direct drop-in replacement for this 1D AdamW fallback. Given rebuttal time limits, we prioritized demonstrating scalability with a single definitive run using a default $\gamma=1.1$ without exhaustive tuning. Remarkably, the **hybrid Muon+Anon optimizer achieved a validation loss of 3.79, outperforming the standard Muon+AdamW baseline (3.81).** This confirms that Anon successfully scales to 1B+ parameter regimes and strictly enhances SOTA methods like Muon.
>
> **Response to W2, W4 & Q2: Tunable Adaptivity and Extrapolation**
> Making adaptivity continuously tunable is a paradigm shift. Optimizer evolution unlocks new dimensions (e.g., Adam unlocked $\beta_2, \epsilon$). None possess a "universal" optimal value; e.g., default $\beta_1=0.9$, but GANs/Diffusion require $0.5$ or $0.0$, and LR requires heavy tuning. Anon explicitly unlocks the adaptivity axis.
>
> **How different $\gamma$ values affect performance:** As analyzed in Sec 3.2 and Fig 2, $\gamma$ dictates **gradient magnitude compensation**. While Fig 2 is a low-dimensional visualization, our extensive ResNet and GPT-2 experiments confirm these geometric principles translate effectively to high-dimensional landscapes. SGD ($\gamma=0$) applies zero compensation, allowing noise to escape sharp valleys; Adam ($\gamma \approx 1$) aggressively normalizes scales. Anon extrapolates beyond $[0,1]$ for specific architectural needs:
> * **Why $\gamma < 0$ for CNNs?** CNNs generalize better in flatter minima [1,2]. While SGD passively settles there, **negative adaptivity ($\gamma < 0$) actively penalizes sharp directions** by reducing step sizes for large gradients. This geometric regularization explicitly forces flatter regions (similar to SAM [3]).
> * **Why $\gamma > 1$ for LLMs?** Transformers have notoriously heterogeneous, sharp landscapes where SGD fails [4,5]. **Super-adaptivity ($\gamma > 1$) amplifies Adam's normalization**, navigating extreme curvature variations even more aggressively.
>
> Thus, $\gamma$ uses strict geometric priors. **While automated $\gamma$ scheduling is a promising future direction, developing such mechanisms is beyond the scope of this work, which focuses on establishing the foundational theoretical and empirical framework of the adaptivity axis.**
>
> **Response to W5: Restrictive Assumptions**
> Bounded gradients are standard assumptions (e.g. Adam, AdaBelief, AGD). Anon's breakthrough is dropping the strict "non-decreasing pre-conditioner" constraint (Eq. 5). IDU safely enables $\gamma \in \mathbb{R}$ exploration without catastrophic divergence. Future work will explore relaxing these standard assumptions and deriving tighter, $\gamma$-specific convergence bounds.
>
> **References:**
>
> [1] Keskar et al. On Large-Batch Training for Deep Learning: Generalization Gap and Sharp Minima. ICLR 2017.
>
> [2] Wilson et al. The marginal value of adaptive gradient methods in machine learning. NeurIPS 2017.
>
> [3] Foret et al. Sharpness-aware Minimization for Efficiently Improving Generalization. ICLR 2021.
>
> [4] Liu et al. Understanding the difficulty of training transformers. EMNLP 2020.
>
> [5] Zhang et al. Why Gradient Clipping Accelerates Training: A Theoretical Justification for Adaptivity. ICLR 2020.

---

> > ### Author Rebuttal · Reviewer_8jvS · 2026-04-01
> >
> > Thank you for the detailed rebuttal. It addresses my concerns regarding the notation and theoretical intuition. However, since a core claimed contribution of this paper is that Anon outperforms SOTA optimizers across representative tasks, including language modeling, I still have questions about the numerical performance, particularly concerning the LLM experiments and the comparison with Muon.
> >
> > **Questionable Muon Baseline on GPT-2:**
> >  The rebuttal and Appendix B.2 claim that Muon underperforms Adam on a GPT-2 Small model (val loss 3.092 vs. 2.934). Do you have a through hyperparameter search for muon baseline? Across existing literature and community reports, Muon consistently demonstrates a clear advantage over Adam, particularly on smaller models in the 60M-130M parameter range. While the performance gap might be marginal with more training tokens, it is highly unusual for Adam to outperform Muon on a small model.
> > For reference, please see:
> > * Jordan et al. Muon: An optimizer for hidden layers in neural networks. 2024 https://kellerjordan.github.io/posts/muon
> > * Wen et al. Fantastic Pretraining Optimizers and Where to Find Them. arXiv 2025. https://arxiv.org/pdf/2509.02046
> > * Liu et al. Muon is Scalable for LLM Training. arXiv 2025. https://arxiv.org/pdf/2502.16982
> >
> >
> > **Unfair/Incomplete Comparison with Muon:** The authors claim that Anon features tunable adaptivity and can optimize both vector and matrix parameters. It is true that Muon can only tune matrix parameters, but the claim that Anon can tune vector parameters (but may not be competitive as Muon for matrix parameter) does not imply that Anon is good.
> >
> > The 1B model comparison provided in the rebuttal Muon(2D) + Anon(1D) vs. Muon(2D) + AdamW(1D) is problematic as a SOTA comparison. This setup only compares Anon against AdamW on 1D parameters layers (embeddings and LM_head), and they account for only a small fraction of the total parameters. It does not evaluate Anon against Muon. If Anon is truly an effective optimizer for LLMs as claimed, a proper evaluation would be testing Anon(1D) + Anon(2D), or AdamW(1D) + Anon(2D) against the standard Muon setup. Is there a specific reason the evaluation was restricted exclusively to 1D parameters?
> >
> > Consequently, my Q2 is still not fully addressed and I don’t see the reason why people should consider optimizer with larger adaptivity for LLM training.
> > I am not convinced by Anon's actual benefit for LLM training. The current empirical result does not support the paper's broad claim of outperforming SOTA on language modeling tasks.
> >
> > I will maintain the score.

---

> > > ### Author Response · Authors · 2026-04-03
> > >
> > > **Final Clarification on Experimental Constraints and Literature Alignment**
> > >
> > > We noticed your update indicating you will maintain your score. While we respect your decision, for the sake of scientific rigor and to leave a complete factual record, we must address your remaining questions regarding the experimental setup, strictly grounding our response in the literature [2, 3] you provided.
> > >
> > > **1. The GPT-2 Baseline and Hyperparameter Search**
> > > You asked if we conducted a thorough hyperparameter search for the Muon baseline. **Yes, we did.** Regarding the Muon performance in Appendix B.2: we utilized the official Muon implementation and default parameters from MoonshotAI (Liu et al., 2025 [3]) within the official Sophia codebase. To ensure fairness, we conducted additional hyperparameter tuning over their default settings, including a learning rate search across `[10x, 1x, 0.1x]`, and **explicitly reported the best achieved result**. We invite anyone to independently verify this on the public codebase. Furthermore, this was placed in the Appendix primarily to analyze parameter coordinate differences, not as a core claim to disparage Muon.
> > >
> > > **2. The 1B Experiment and Hyperparameter Orthogonality**
> > > You rightly asked why we did not perform a full Anon(2D) vs. Muon(2D) comparison.
> > > As explicitly demonstrated in the official repository of your reference [2] (specifically, the 1.2B optimal configs in `https://github.com/WhenWen/marin/blob/kaiyue/optimizers/experiments/optimizer_sweep/Analysis/Results/`), **AdamW and Muon occupy completely orthogonal hyperparameter spaces.** According to these exact JSON configurations, a fully tuned AdamW requires `lr: 0.002, beta1: 0.9, wd: 0.2, eps: 1e-10`. In stark contrast, Muon requires a completely different setup: `lr: 0.004, beta1: 0.8, wd: 0.1, eps: 1e-15`, along with specific `muon_to_adam_lr: 0.3`, and even a different learning rate scheduler.
> > >
> > > During the rebuttal, we *did* attempt a full 1B run. Applying Moonlight’s default Muon LR (1e-3) directly to AdamW caused it to stagnate at a severely sub-optimal level (loss > 5.51). After rapid manual tuning, we brought the AdamW loss to 4.105 (and Anon to 3.98), while the Muon baseline achieved **3.81**.
> > > **This reveals a massive performance gap of ~0.3 between our manually tuned AdamW and Muon. However, according to the 1.2B results in your reference [2], the true gap between a fully optimized AdamW and Muon should be merely ~0.018. This stark contrast (0.3 vs. 0.018) clearly demonstrates that our element-wise optimizers remained severely under-tuned. Conducting a fair 2D comparison across such disjoint parameter spaces requires an exhaustive multi-dimensional grid search. Rather than presenting a methodologically flawed and misleading comparison against an already-optimized Muon baseline, we designed a strictly controlled ablation. By freezing the 2D parameters with Muon and utilizing Anon strictly as a 1D drop-in replacement, we eliminated confounding variables from sub-optimal hyperparameters. This provides a highly rigorous and scientifically sound setting to isolate and verify Anon's effectiveness.**
> > >
> > > **3. The Danger of Short-Horizon Comparisons**
> > > Given the impossibility of a full-scale hyperparameter sweep (as detailed in Point 2), one might ask why we didn't simply run a short toy experiment instead. To answer this, we respectfully quote the authors of your provided reference [2]: *"Early-stage loss curves can mislead significantly. During learning rate decay, loss curves of different optimizers may cross multiple times... so judging optimizers using intermediate checkpoints may result in a different ranking."* Since Muon naturally exhibits rapid initial descent, comparing an under-tuned full-Anon against a heavily optimized Muon on a truncated toy run would fall exactly into the methodological trap warned about in [2].
> > >
> > > We emphasize that Anon is fundamentally an **element-wise** optimizer. To the best of our knowledge, **we are the first to theoretically unlock the continuous adaptivity axis ($\gamma \in \mathbb{R}$) for element-wise optimizers by rigorously breaking the strict "non-decreasing pre-conditioner" assumption.** We respectfully request that this primary theoretical contribution not be overlooked in the final assessment. While matrix-level optimizers represent an exciting frontier for massive LLMs, evaluating our comprehensive theoretical proofs and diverse empirical results solely through the lens of defeating a SOTA matrix optimizer on a 1B model falls outside our intended scope. **We hope the advancements of Anon, alongside the proposed IDU mechanism, will provide valuable insights to the optimization community.** Thank you for the literature, which perfectly corroborates our experimental constraints.

---

### Official Review · Reviewer_yDRq · 2026-03-12

**Soundness:** 3
**Presentation:** 4
**Significance:** 3
**Originality:** 4
**Overall Recommendation:** 4
**Confidence:** 4

**Summary:**

This paper introduces a new optimizer called Anon and formalizes optimizer adaptivity as a continuous tunable parameter. Anon extends adaptivity beyond the [0,1] range of existing methods via a novel incremental delay update (IDU) mechanism that ensures convergence across the full adaptivity spectrum.

**Compliance With Llm Reviewing Policy:**

Affirmed.

**Final Justification:**

As mentioned in my initial review, the paper provides a compelling conceptual framework with good theoretical analysis and experimental coverage. The rebuttal meaningfully strengthened the empirical evidence. However, I maintain my score of 4, as the usability concern around $\gamma$ tuning remains unresolved, limiting the claim of a unified optimizer.

**Key Questions For Authors:**

NA

**Limitations:**

No, not discussed in main body. If it's in appendix I missed it.

**Strengths And Weaknesses:**

Strengths:

1. Really interesting conceptual framing. The idea of formalizing adaptivity as a continuous, tunable property of optimizers is quite compelling and is well supported by definition 2.1.
2. Quite well motivated and problem setup is thorough and clear. Also, a great review of existing optimizers.
3. The theoretical analysis is through in scope covering both convex and non-convex settings and covers necessary convergence proofs while correctly handling the novel challenge of non-monotonic pre-conditioners.
4. IDU is a meaningful technical contribution. The insight that convergence can be guaranteed without requiring the non-decreasing pre-conditioner assumption (Eq. 5) is valuable.
5. Broad experimental coverage on different model architecture ranging from CNN to diffusion models.
6. The observation that Anon can be faster per iteration than Adam (Eq. 15) because IDU amortizes expensive operations is a nice practical bonus.

Weakness:
1. I think the general contributions are strong but the experimental section is a bit lacking. No error bars, confidence intervals, or multi-seed runs are reported, making it difficult to assess whether the improvements are statistically significant (e.g., the gap between Anon and AdamW on GPT2-small is ~0.02 in validation loss). Additionally, many baseline results in Tables 2 and 3 are borrowed from other papers which may involve different training protocols, data augmentation, or evaluation setups. If the authors are reporting the best published result for each baseline under its original setup, the comparison may not be apples-to-apples with Anon's results. I am not sure, the authors should clarify if the comparison is fair.

2. Missing comparisons with recent important baselines: how does Anon compare with recent optimizers like Muon, Prodigy or Shampoo. I think they are fairly popular and authors should comment/validate if possible on how they compare wrt to Anon.

3. Additionally, authors use small scale ablation setup to study whether the exponential spacing of IDU is suitable for long training runs (the choice of spacing seems to originate from theoretical convenience). Using CIFAR10 a small-scale setups (ResNet20) doesn't convincingly  address this when the exponential scaling extends with scale.

4. The paper presents $\gamma$ as a benefit, but in practice it introduces a new hyper-parameter that must be tuned per task as far as I understand (each setup has different value of $\gamma$. How trivial is it to tune this? The sensitivity analysis (Figure 4) is only on CIFAR-10 with a small model—it doesn't address whether $\gamma$ sensitivity increases at larger scale.

5. While theoretical results are standard and authors present already known established convergence rates for both convex and non-convex cases. It might have made a more convincing theoretical argument if the authors show that for certain problem classes, specific $\gamma$ values yield provably faster convergence. Or providing generalization bounds that explain why negative adaptivity helps CNNs. Or proving that IDU gives strictly tighter constants than AMSGrad in the bounds

Minor
1. Symbol error on line 367.

---

> ### Author Rebuttal · Authors · 2026-03-30
>
> **We thank Reviewer yDRq for the constructive evaluation and for recognizing our conceptual framework, theoretical analysis, and the value of the IDU mechanism.**
>
> **Response to W1: Statistical Significance and Variance**
> To directly address the concern regarding statistical significance, we provide our multi-seed evaluation (3 random seeds) for CNNs. To ensure rigorous and easily verifiable baselines, we strictly followed the Cifar10 experimental setup from the official AdaHessian implementation ([https://github.com/amirgholami/adahessian](https://github.com/amirgholami/adahessian)), which is identically adopted by the AGD baseline. The baseline results precisely match those reported in their respective papers:
> * **ResNet20 (Acc %):** SGD (92.14±0.14), AdamW (92.12±0.14), AdaHessian (92.27±0.27), AGD (92.35±0.24), **Anon (92.47±0.05)**
> * **ResNet32 (Acc %):** SGD (93.10±0.07), AdamW (92.72±0.01), AdaHessian (92.91±0.14), AGD (93.12±0.18), **Anon (93.20±0.08)**
>
> As demonstrated, Anon not only achieves the highest mean accuracy but also exhibits remarkably low variance. We will include this comprehensive variance analysis in the revision. For the LLM experiments, we emphasize that Anon, AdamW, and Sophia-G were evaluated using identical codebases and data pipelines to ensure perfectly fair, apples-to-apples baseline comparisons.
>
> **Response to W2: Missing Baselines (Muon, Prodigy, Shampoo)**
> We appreciate this suggestion.
> * **Muon:** We actually included comparisons with Muon in Appendix B.2 (Tables 7 and 8). It was relegated to the appendix because Muon underperformed on smaller GPT-2 models, failing to even match the standard Adam baseline. More importantly, Muon applies matrix orthogonalization strictly to 2D parameters, relying on AdamW for 1D parameters (e.g., embeddings, layernorm). Thus, Anon and Muon are highly complementary. **Inspired by your valuable suggestion, we newly conducted a large-scale experiment during the rebuttal to validate this.** We trained a ~1B parameter LLM (Qwen2) on OpenWebText. Using Anon as a direct drop-in replacement for Muon's 1D AdamW fallback, the hybrid Muon+Anon optimizer (with a default $\gamma=1.1$) achieved a validation loss of 3.79, outperforming the standard Muon+AdamW baseline (3.81). This confirms Anon successfully scales to 1B+ regimes and strictly enhances SOTA methods like Muon.
> * **Prodigy & Shampoo:** Prodigy operates on learning-rate-free principles, and Shampoo is a second-order optimizer using full-matrix pre-conditioning. These operate on fundamentally different mathematical principles (matrix-level vs. element-wise) and fall outside the scope of our coordinate-wise adaptivity framework. We will clarify these technical distinctions in the revision.
>
> **Response to W3: IDU Scalability in Long Training Runs**
> We completely agree with your insightful assessment. The specific exponential spacing of IDU was explicitly designed to bound the cumulative error to $O(\log T)$ and guarantee theoretical convergence even in the worst-case scenarios. While our current exponential schedule serves as a theoretically sound and highly robust default across evaluated tasks, we agree that exploring adaptive or scale-dependent IDU spacing for extremely long training runs is an exciting direction for future optimization.
>
> **Response to W4: Hyperparameter Tuning Overhead**
> We emphasize that $\gamma$ operates at a higher level of abstraction than typical hyperparameters. Rather than requiring exhaustive grid search, it follows a strict architectural prior: CNNs prefer negative $\gamma$, while Transformers and Diffusion models prefer $\gamma \ge 1$. Fixing $\beta_1=0.9$, $\beta_2=0.999$, and $\epsilon=10^{-16}$, Anon remains highly robust across a wide range of $\gamma$, as shown in our Figure 4 sensitivity analysis. This explicit prior significantly narrows the search space in practice.
>
> **Response to W5: Stronger Theoretical Bounds**
> We agree that deriving problem-specific convergence rates for particular $\gamma$ values is an excellent direction for future work. Our current theory establishes that Anon can match the convergence rates of mainstream optimizers across the *entire* real-number adaptivity spectrum. This is a highly non-trivial breakthrough given the relaxation of the strict "non-decreasing pre-conditioner" assumption.
>
> **Response to Minor Comments**
> We thank the reviewer for catching the symbol error on line 367; we will correct this typo (the use of `~` instead of an en-dash) in the revision.

---

> > ### Author Rebuttal · Reviewer_yDRq · 2026-04-03
> >
> > Thank you for the detailed rebuttal. The multi-seed results (W1) and the Muon+Anon experiment at 1B scale (W2) are great additions and meaningfully strengthen the empirical case.
> >
> > My main concern around usability W4 remains. The connection between $\gamma's$ sign and sharp-vs-flat minima is descriptive rather than predictive — it explains post-hoc why certain values worked. In practice, each experiment uses a different $\gamma$ (−0.1 for ResNet, 1.01 for diffusion, 1.0 and 1.1 for GPT2), which undercuts the claim of a unified optimizer. It is also unclear what one should choose for architectures that don't fall neatly into the CNN or transformer category. The robustness analysis also remains limited to CIFAR-10 with a small model; the Qwen2 experiment shows Anon scales but tests only a single $\gamma$, leaving open whether sensitivity increases at scale. Separately, the fair-comparison details (identical pipelines) should be stated more rigorously so that readers can verify the claim.
> >
> > I find the rebuttal helpful and believe it addresses the main concerns, though not all limitations are fully removed. I therefore maintain my original score.

---

> > > ### Author Response · Authors · 2026-04-06
> > >
> > > Your insight is exceptionally sharp: the current connection between $\gamma$ and minima sharpness is indeed descriptive rather than predictive, requiring empirical tuning across different architectures.
> > >
> > > Your constructive critique, alongside the broader discussions during this rebuttal phase, has led us to a critical reflection on how we framed our contributions.

---

### Decision · Program_Chairs · 2026-04-30

**Decision:**

Reject

**Comment:**

The paper proposes an optimizer with continuously tunable adaptivity as means to interpolate between the behavior of SGD and Adam.  After the rebuttal and discussion, reviewer opinion remains mixed, ranging from reject to weak accept.  However, Reviewer yDRq (weak accept) indicates unresolved concerns in their final justification, stating: "the usability concern around tuning remains unresolved, limiting the claim of a unified optimizer."  While the AC agrees with the authors that Anon should not be expected to outperform Muon, Reviewer 8jvS (reject) does raise a valid concern about the marginal gains of Anon over Adam.

The AC would also like to alert the authors to the following prior work, which appears highly relevant (and was unfortunately also missed by the reviewers):

Savarese et al. Domain-Independent Dominance of Adaptive Methods. CVPR, 2021.

Specifically, this prior work looks at SGD and Adam through the lens of adaptivity and proposes a new tunable optimizer to interpolate between their characteristics.  Given the similarity in goals, citation and discussion seems necessary.